

# Extreme water levels, waves and coastal impacts during a severe tropical cyclone in Northeast Australia: a case study for cross-sector data sharing

Thomas R. Mortlock[1,2], Daryl Metters[3], Joshua Soderholm[4], John Maher[3], Serena B. Lee[5], Geoffrey Boughton[6], Nigel Stewart[4], Elisa Zavadil[7], Ian D. Goodwin[2]

[1]Risk Frontiers, St Leonards, 2065, Australia
[2]Department of Environmental Sciences, Macquarie University, North Ryde, 2109, Australia
[3]Coastal Impacts Unit, Department of Environment and Science Queensland Government, Deagon, 4017, Australia
[4]Fugro Roames, Runcorn, 4113, Australia
[5]Griffith Centre for Coastal Management, Griffith University, Gold Coast 4215, Australia
[6]Cyclone Testing Station, James Cook University, Douglas, 4811, Australia
[7]Alluvium, Cremorne, 3121, Australia

*Correspondence to*: Thomas R. Mortlock (thomas.mortlock@riskfrontiers.com)

**Abstract.** Severe Tropical Cyclone (TC) *Debbie* made landfall on the north Queensland coast of Australia on 27 March 2017 after crossing the Great Barrier Reef as a slow-moving Category 4 system. Groups from industry, government and academia collected coastal hazard and impact data before, during and after the event and shared this data to produce a holistic picture of TC *Debbie* at the coast. Results showed the still water level exceeded the highest astronomical tide by almost a metre. Waves added a further 16 percent to water levels along the open coast, and were probably unprecedented for this area since monitoring began. In most places, coastal barriers were not breached and as a result there was net offshore sand transport. If landfall had occurred two hours earlier with the high tide, widespread inundation and overwash would have ensued. This paper provides a case study of effective cross-sector data sharing in a natural hazard context. It advocates for a shared information platform for coastal extremes in Australia to help improve the understanding and prediction of TC-related coastal hazards in the future.

## 1 Introduction

Storm tide and wave impacts associated with tropical cyclones (TC) can result in significant loss of life, damage to coastal infrastructure and property. The combination of wind setup with a barometric surge, resulting from low atmospheric pressure, can elevate water levels above the astronomical tide and breach coastal barriers and defences. The presence of breaking waves can further increase the potential for inundation through wave setup and runup, and cause coastal erosion and structural damage. In many cases, storm tide and wave impacts coincide with pluvial and fluvial flooding during TCs, which can further elevate water levels locally and produce complex and damaging hydrodynamic conditions where river systems meet the ocean.

The Australian region has high exposure to TCs, with an average of 12 events occurring per year with approximately 5 making landfall (between 1961 and 2017, BoM, 2017). They mainly affect the northern coastline from central Queensland on the east coast (South Pacific Ocean) to the northwest coast of Western Australia (Indian Ocean), although the extra-tropical transition of some TCs to Tropical Lows means impacts can be felt further south (Haigh et al., 2014). The highest storm tides (100-year return period levels > 4m) occur on the northwest (Western Australia) coast, while along the northeast (Queensland) coastline storm tides are typically between 2 and 4 m (McInnes et al., 2016). Some paleoclimate indicators, however, suggests storm tides have exceeded this range on the Queensland coast during the most extreme events over the Holocene (Nott, 2015).

The magnitude of TC-induced storm surge is not linearly related to cyclone intensity. Shoreface slope, shoreline geometry, wind obliquity to the coast, radius of maximum winds (RMW), cyclone track and forward-moving speed all contribute to the surge-producing potential. The existence of offshore islands and reefs can further modulate hydrodynamic conditions to the lee of these



barriers, either amplifying or dampening water levels (Lipari et al., 2008). In general, surge potential is maximised on open, straight coastlines with shallow shoreface slopes and slow-moving, landfalling cyclones travelling perpendicular to the coast. In the Southern Hemisphere and on east-facing coasts, the clockwise flow of low pressure systems means coastal areas on the southern limb of TCs experience the most wind, surge and wave impacts. Conversely, areas to the north often experience a suppression (set-
down) of water levels and low wave impacts due to offshore-directed winds.

TC-induced coastal flooding is maximised when storm surge coincides with a high astronomical tide and energetic wind-waves. Non-linear harmonics between tide and surge can further elevate total water levels especially in areas with large tidal ranges, with some research suggesting surge maxima are more likely to occur on the rising or falling tide rather than at slack water because of this dynamical coupling (Horsburgh and Wilson, 2007). Wave breaking further increases water levels at the coast (wave setup)
and adds more forward momentum to the water mass (wave runup) meaning coastal foredunes can be breached even when the storm tide elevation is lower than the dune crest.

The TC season in Australia typically runs from November through to April (Austral late summer to early autumn), when sea surface temperatures are warm enough for cyclogenesis. In Australia, as elsewhere, TC-affected coasts (< 28 ° S) comprise tide-modified or tide-dominated beach systems with meso- to macro-tidal range (2 to 4 m, or exceeding 4 m), low-relief dunes and a
steep high-tide beach fronted by shallow gradient sand, rock or reef flats (Short, 2006). These beaches are equilibrated with a predominantly low-energy hydrodynamic regime, punctuated by infrequent high-energy TC events. The large energy difference between the modal and extreme regimes leaves exposed tropical-coast locations vulnerable to significant erosion during TCs. This can have lasting impacts for beach amenity, access and tourism, and lowers the geomorphic threshold for subsequent inundation and erosion.

On 28 March 2017, a Severe Tropical Cyclone (*Debbie*) made landfall on the central north Queensland coast of Australia. *Debbie* affected the Whitsunday Islands group, approximately 30 km offshore, as a low-end Category 4 tropical cyclone before making landfall on the mainland coast, near Airlie Beach, as a high-end Category 3 tropical cyclone (BoM, 2018). This area has experienced several TCs in recent years - although not as intense as *Debbie* - including *Ului* in 2010 (category 3), *Anthony* in 2011 (category 2), and *Dylan* (category 2) in 2014. *Debbie* intensified from a tropical low southeast of Papua New Guinea in the Coral
Sea and proceeded to drift south as a category 2 system. Three days before crossing the coast, *Debbie* turned southwest (perpendicular to the coast) and rapidly intensified to category 4 with a RMW of approximately 30 km, reducing to 15 km at landfall (BoM, 2017).

Because of the intense winds (peak gusts over 260 km h$^{-1}$), shore-normal approach, slow forward-moving speed and coincidence with rising spring tides, significant storm tide inundation was expected. *Debbie's* track on approach to the coast and landfall
location was in fact very similar to Severe TC *Ada* in 1970, an infamous category 4 system which was responsible for the loss of 14 lives. It eventuated that the slowing of *Debbie* on approach to the coast (to only 7 km h$^{-1}$ shortly after landfall) meant that landfall occurred approximately two hours after high tide, avoiding more substantial and widespread flooding. However, significant coastal impacts were still experienced at certain locations south of the landfall site, resulting from a combination of localised hydrodynamic processes and regional coastal geometry – as discussed further in this paper.

Approximately 15 hours after crossing the coast, *Debbie* weakened to a tropical low but continued to cause significant wind and flood damage and dangerous coastal conditions throughout southeast Queensland, northern New South Wales and subsequently travelled across the Tasman Sea to impact New Zealand. In total, *Debbie* resulted in almost AUD $1.7 billion insured losses (PERILS, 2017), making it the most expensive global natural disaster in the first half of 2017.

Despite the significant and lasting impacts of tropical cyclones on the coast, there is a lack of observational data to support process
knowledge and constrain coastal hazard modelling of these events in Australia. By comparison, the impacts of extra-tropical





cyclones on coastal systems is much better understood (e.g. Turner et al., 2016, Strauss et al., 2017). To address this data gap, this paper collates and analyses observations of coastal impacts and concurrent hydrodynamic conditions before, during and directly after severe tropical cyclone *Debbie*. Field data were collected independently by a group of organisations (Risk Frontiers/Macquarie University, Fugro Roames, Department of Environment and Science (DES) Queensland Government, James

Cook University, Griffith University and Alluvium), using a range of data collection and analysis methods.

This work represents the only observational analysis of hydrodynamic drivers and concurrent coastal impacts for a severe tropical cyclone in Australia. The data is of value for the calibration and validation of coastal hazard modelling, both in north Queensland and for other tropical cyclone-affected east coasts in the Southern Hemisphere, such as Mozambique, Tanzania and Brazil, where the genesis of these events are similar, but coastal observations are lacking. This paper also provides a case study in data sharing

and open collaboration across industry, government and academia, something which is currently lacking in a natural hazard risk context in Australia.

## 2 Study area

*Debbie* made landfall near Airlie Beach (20.3 °S, 148.7 °E), on the Whitsunday (central north) Queensland coast, around 12:40 pm on 28 March 2017. Prior to landfall, *Debbie* travelled south-west from the Coral Sea over the southern end of the Great Barrier

Reef (GBR), the Whitsunday Island group, and then inland as a tropical low before turning southeast towards Brisbane, the capital of the state of Queensland (Figure 1). The Whitsunday and Mackay regions, from Airlie Beach in the north to Mackay in the south (~110 km shoreline length), were the focus for the groups' surveys as these were the areas within the influence of *Debbie*'s southern limb and onshore-directed flow.

20         [Figure 1 here]

### 2.1 Wave and hydrodynamic regime

Most beaches on the central-north coast of Queensland receive little or no ocean swell waves and are only exposed to low and short-period wind waves generated primarily by south-easterly trade winds. The median long-term significant wave height, $H_s$, at the Mackay wave buoy (1975 - 2017) is 0.7 m, peak spectral wave period, $T_p$, is 5.8 s, and mean wave direction, *MWD*, (2002 -

2017) is 115 degrees True North (° TN). During cyclone events, conditions greatly exceed this, with wave heights greater than 5 m not uncommon. The predominately low wave energy regime is a result of effective attenuation of ocean swells by the GBR matrix, which means nearshore wave conditions are largely dependent on local wind speeds rather than far-field wave generation (Gallop et al., 2014).

The Whitsunday and Mackay coastline is situated to the lee of the widest section of the GBR shelf, with over 180 km separating

the outer reefs and the mainland coast between Bowen and Mackay. In this area, water depths do not exceed 80 m but the bathymetry is highly variable. The mean spring range at Hay Point (~20 km south of Mackay) is 4.9 m, whereas on narrower sections of the GBR, typical spring ranges are around 2 to 3 m.

### 2.2 Geomorphic setting

The tidal range at Mackay is seven times greater than the mean annual wave height (at the Mackay buoy, 35 m water depth). For

this reason, beaches in this area are tide-dominated to tide-modified (Short, 2000), with a relatively steep high-tide beach and wide,

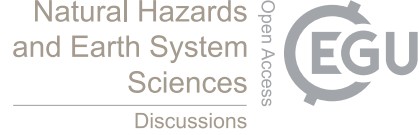

very low gradient sand and/or tidal mud flats. At low tide, the water line may be several hundred metres seaward of the high-tide beach. Tidal state therefore plays an important role in modulating storm surge impact in this region.

Coastal embayments in north Queensland are typically east-orientated, grading from a protected southern end dominated by tidal flats and wind-blown beach ridges, towards a more exposed northern end oriented perpendicular to the south-easterly trade winds and backed by transgressive sand dunes (Short, 2000). Longshore transport is generally northward, but can change direction in the lee of headlands.

### 2.3 Study sites

Results from seven coastal sites are presented. These include (from north to south) Airlie Beach (20.3 °S, 148.7 °E), Hamilton Island (20.4 °S, 149.0 °E), Wilson Beach (20.5 °S, 148.7 °E), Conway Beach (20.5 °S, 148.8 °E), Laguna Quays (20.6 °S, 148.7 °E), Midgeton Beach/Midge Point (20.6 °S, 148.7 °E) and Seaforth Beach (20.9 °S, 148.9 °E).

Observations of beach morphological change and storm demand are presented at the main study sites of Wilson Beach, Conway Beach and Midgeton Beach, which were some of the most severely impacted coastal locations during *Debbie*.

Wilson Beach is a small coastal settlement with a 300 m-long beach that faces south across the sand flats and channel of the Proserpine River. The shore consists of a steep high tide beach, fronted by 200-m wide sand and mud flats, adjacent to the river channel (Short, 2000). The houses located closest to the shoreline are built on the sand foredune ridge, with as little as 30 m separating the front doors from the beach crest, which has an elevation of around 5 m Australian Height Datum (AHD).

Conway Beach is a small residential settlement located 2 km south-east of Wilson Beach, with a 1.5-km long beach facing south-east across the Proserpine River mouth. The seafront has a low gradient high-tide beach flattening to 400 to 500 m wide, low sand flats (Short, 2000). The western portion of the beach (in front of the settlement) is backed by a rock revetment with a crest of between 5 to 6 m AHD. The houses located closest to the shoreline sit approximately 50 m behind the revetment. The eastern section of the beach is backed by a vegetated coastal dune and creek system.

Midgeton Beach is a 1.8-km long, south-east facing, low gradient, sandy beach. The high-tide beach is fronted by a wide, low gradient intertidal beach, with sand flats extending up to 1 km off the southern end in front of the Yard Creek mouth, which forms the southern boundary. A smaller, mangrove-fringed creek (Sandfly Creek) forms the northern boundary (Short, 2000). The community is situated along the northern half of the beach. Foreshore houses initially sit approximately 70 m behind the sparsely vegetated low-lying foredune. Moving southward, the land between the houses and foredune becomes increasingly vegetated and to the south of the community the beach is backed by a foreshore reserve fringed with dense vegetation and coconut palms.

Observations of inundation limits are presented for Airlie Beach, Midge Point, Laguna Quays and Seaforth Beach. Observations of interior and structural damage to buildings resulting from storm tide and waves are presented for Wilsons Beach, Conway Beach and Hamilton Island. The locations of all sites are shown in Figure 1.

### 3 Datasets and methods

### 3.1 Storm tide gauges

Storm tide monitoring in Queensland was initiated in the mid-1970s and comprises a network of 36 gauges, which all now measure water levels every minute, relative to local Lowest Astronomical Tide (LAT). In this study, elevations were converted to AHD (Australian Height Datum), which approximates to mean sea level. All storm tide gauges are fitted with a barometer which records the atmospheric pressure at the gauge in parallel with water level.



We define 'storm tide' in this paper as the astronomical tide plus meteorological surge components (wind setup plus barometric surge) but not wave breaking effects which are treated separately. Observations from four storm tide gauges located within a 150 km radius of *Debbie's* landfall site were analysed. These include (north to south); Bowen (located on the main cargo wharf at Bowen), Shute Harbour (at Shute Harbour Wharf), Laguna Quays (at Laguna Quays Marina), and Mackay (on Mackay Outer Harbour, Pier 1) (Table 1, locations Figure 1).

[Table 1 here]

### 3.2 Barometric surge

The meteorological storm surge during TC events includes wind setup and an accompanying barometric component. Under strong, onshore-directed wind forcing, the water surface is tilted upward with distance downwind, causing an increase in the water level towards the coast (wind setup) (Kamphuis, 2010). A barometric surge occurs when there is a surface air pressure difference between the sea and shore. The inverse barometer (IB) effect can be approximated as;

$$\Delta h = 10^2 \left( \frac{1}{\rho g} P - P_{ref} \right) \tag{1}$$

Where $\Delta h$ is the change in the sea surface height at the shore (m), $\rho$ is the density of seawater (1025 kg m$^3$), $g$ is the acceleration due to gravity (9.81 m s$^{-1}$), $P$ is the central atmospheric pressure of the cyclone and $P_{ref}$ is the global mean atmospheric pressure (1013.3 hPa). $P$ was taken as the minimum barometric recordings at each storm tide gauge. The effect of $1/\rho g$ is small (approximately 0.01 m), such that Eq. (1) is often reduced to $P_{ref} - P$.

To estimate wind setup, the IB effect (Eq. (1)) was subtracted from the residual between the predicted astronomical tide and the observed water level. In reality, the residual may also include errors in the harmonic derivation of the astronomical tide and non-linear tide-surge interactions (Horsburgh and Wilson, 2007). However, the contribution of these components is outside the scope of this study, so the 'residual' is taken as synonymous with the meteorological storm surge.

### 3.3 Wave buoys

Ocean wave monitoring in Queensland began in the mid-1970s and now comprises a network of 16 waverider buoys. Observations from one buoy located to the north of the landfall site, Abbot Point (14 m water depth), and two to the south, Mackay (34 m) and Hay Point (10 m), were used in this paper (Table 1, locations Figure 1).

The buoys measure directional wave spectra, from which parametric data are derived at 30-minute intervals. In this study, we refer to the 0.5-hrly significant wave height, $H_s$ (m), the maximum wave height, $H_{max}$ (m), the peak spectral period, $T_p$ (s), and the mean wave direction, $MWD$ (° TN, degrees True North). $H_s$ approximates to the average of the highest third of all waves measured in 30 minutes and is a common descriptor of the bulk wave climate; $H_{max}$ is the maximum wave height recorded during a half-hour; $T_p$ is the wave period pertaining to the primary energy peak of the wave spectrum (the most energetic waves in 30 minutes); and $MWD$ is the mean wave direction at the primary energy peak.

### 3.4 Wave data extrapolation

Due to extreme conditions leading up to *Debbie*, the Mackay buoy failed at around 05:30 on 28 March (~ 7 hrs before landfall). Prior to failure, a strong correlation was observed between wave conditions at Mackay and Hay Point ($R$ 0.82, $p \leq 0.05$ for $H_s$),





located only ~ 35 km to the south. This suggests missing data at Mackay can be estimated by extrapolating observations from Hay Point.

To do this, a cumulative distribution function (CDF) mapping approach was used (Brocca et al., 2011). This method compares the Mackay and Hay Point CDFs and adjusts the Hay Point CDF to best match the Mackay CDF. The Mackay-adjusted Hay Point

data was then used as a surrogate for the missing data at Mackay. Figure 2 illustrates this process for extrapolating $H_{max}$; this was repeated for $H_s$, $T_p$ and $MWD$.

[Figure 2 here]

### 3.5 Wave runup estimation

Water levels at open-coast sites include the effects of swash, wave-induced setup and runup, in addition to the storm tide. Swash is generally defined as the time-varying location of the intersection between the ocean and the beach. Wave-induced setup is the super-elevation of the still water level due to the presence of waves. Wave runup is the maximum vertical extent of wave uprush on a beach or structure. Most calculations of runup include the effects of swash and wave-induced setup, and are therefore a measure of the maximum elevation of wave influence above the still water level (or in this case, storm tide). It follows, therefore,

that the maximum inundation extent - as evidenced by the most landward detritus lines observed in the field – represents the elevation reached by wave runup above the storm tide.

Wave runup was estimated empirically from wave buoy and beach profile observations, using the equation of Stockdon et al. (2006):

$$R_{2\%} = 1.1 \left( 0.35 \tan\beta \left( H_{s0} L_0 \right)^{0.5} + \frac{H_{s0} L_0 \left( 0.563 \tan\beta^2 + 0.004 \right)^{0.5}}{2} \right) \tag{2}$$

Where $R_{2\%}$ is the elevation above the storm tide level that is exceeded by 2 % of the wave runups, $\tan\beta$ is the beach slope, and $H_{s0}$ and $L_0$ are the offshore (i.e. deepwater, where water depth, $d \geq 0.5\,L$) significant wave height and wave length, respectively. This equation has been applied here because it is valid for a broad range of sandy beach types and has been applied for Hurricane-type

wave conditions (Stockdon et al., 2007).

We used $H_s$ at the time of maximum storm tide (not the peak-storm $H_s$) to calculate $R_{2\%}$, since the storm tide component accounts for most of the total water level and thus represents the assumed time of maximum inundation. $H_s$ was taken from the Mackay buoy, to the south of *Debbie* (wave conditions to the north were offshore-directed, thus not contributing to wave runup). The wavelength at the time of the maximum storm tide was first estimated:

$$L = \frac{gT_p^2}{2\pi} \tanh\left( \frac{2\pi d}{L} \right) \tag{3}$$

From this, wave conditions at the time of the peak storm tide were shown to be in intermediate water depths ($0.05\,L \leq d \leq 0.5\,L$) at the buoy (35 m depth), thus not satisfying the deep-water condition for Eq. (2). $H_s$ was thus de-shoaled to a deepwater value

using linear wave theory:





$$H_{s0} = \sqrt{\frac{c_g}{c_{g0}}} H_s \qquad (4)$$

Where $c_g$ and $c_{g0}$ are the wave group speeds at the buoy and in deepwater, respectively, given as:

$$c_g = 0.5 \left[ 1 + \frac{4\pi d/L}{\sinh(4\pi d/L)} \right] \left[ \frac{gT_p}{2\pi} \tanh\left( \frac{2\pi d}{L} \right) \right] \qquad (5)$$

$$c_{g0} = \frac{gT_p}{4\pi} \qquad (6)$$

This resulted in an $H_{s0}$ of 5.0 m and $L_0$ of 131 m at the time of peak storm tide, which was used to estimate $R_{2\%}$ at study sites on

the south side of *Debbie* ($L_0$ is equal to $L$ in Eq. (3) as $T_p$ is assumed not to change during shoaling).

The beach slope used in Eq. (2) is defined over the area of significant swash activity (Stockdon et al., 2006). However, for applications to cyclone-induced runup where large waves likely move the swash zone higher up the beach profile, the upper (high-tide) beach slope ($\tan\beta$) is a more relevant measure (Stockdon et al., 2007). $\tan\beta$ was estimated for each study site, from cross-shore beach profiles, from the break of slope at the toe of the high-tide beach up to the berm crest or toe of the coastal defence

(where present).

### 3.6 Wave power estimation

Wave power is a measure of the energy flux potential in a wave of a given height travelling at a given speed. The maximum power of waves generated during tropical cyclones is an important statistic for the design of coastal and offshore structures. Likewise, the cumulative power of waves during a TC event is an important indicator of beach and dune erosion potential (Splinter et al., 2014).

The deepwater wave power, $P_0$, can be estimated from:

$$P_0 = \frac{1}{16} \rho g H_{s0}^2 c_g \qquad (7)$$

From this, the cumulative storm wave power, $P_c$, integrated over the duration of the storm, $D$, is:

$$P_c = \int_0^D P_0 dt \qquad (8)$$

Where $P_0$ and $P_c$ are expressed in megawatt hours per metre crest-length (mWh m). Storm wave conditions were defined as those exceeding a threshold $H_s$ of 2 m, which approximates to the 95% percentile long-term wave height at Mackay, a threshold that

conforms with other studies of storm waves in East Australia (e.g. Splinter et al., 2014, Goodwin et al., 2016).



### 3.7 Airborne terrestrial LiDAR surveys

Fugro Roames undertook aerial LiDAR (Light Detection And Ranging) surveys eight months prior to *Debbie* (23 July - 3 August 2016) and four to five days after *Debbie* (1 - 2 April 2017), at locations across Mackay and the Whitsundays. The primary purpose was to provide a rapid assessment of energy distribution networks, but flight paths also covered coastal sections of interest to this

study. Data were collected from an altitude of approximately 1850 ft (564 m), with a track spacing of 350 m. Reported vertical accuracy was ± 0.15 m (to one RMSE, or 68% confidence). Because the LiDAR surveys were flown for terrestrial applications, data were only available landward of the waterline. The Digital Elevation Models (DEMs) interpolated from the LiDAR points were processed to remove vegetation and buildings.

### 3.8 Beach profile and surge limit surveys

Cross-shore beach profiles were taken through the LiDAR DEMs pre- and post-*Debbie*. In addition, Differential GPS (DGPS) surveys were undertaken by Risk Frontiers/Macquarie University (RF/MQU) along the same profile lines approximately five months after *Debbie*. DGPS measurements of estimated maximum inundation extent were also made directly after *Debbie* by RF/MQU and DSITI. These measurements were largely based on observed debris lines (usually pumice, vegetation or coral). An on-the-fly GNSS system with post-processing was used for the RF/MQU DGPS surveys. The mean vertical accuracy was ±

0.33 m (to one RMSE, or 68% confidence), because of not being able to calibrate against a permanent survey marker (PSM). For this reason, more data points were recorded over the same locations to increase the precision of the mean value. DSITI used Real-Time Kinematic (RTK) GNSS with post-processing. At each location, calibration points were taken on PSMs adjacent to the work area, giving a better (mean) vertical accuracy of ± 0.04 m.

### 3.9 Structural damage surveys

A team from the Cyclone Testing Station at James Cook University collected land-based and geo-tagged photographic evidence of damage caused to buildings in the areas affected by storm tide inundation and waves. These surveys were focussed on Wilson Beach and Hamilton Island as the two locations receiving the most water damage to buildings.

### 4 Results

### 4.1 Storm tide and surge conditions

Figure 3 shows the water level variations and surge residuals at the four gauges before and after landfall (12:40 on 28 March).

[Figure 3 here]

### 4.2 Wind setup and barometric surge

Using Eq. (1), the IB and wind setup contributions to the total surge were estimated at each gauge (Table 2). Figure 4 shows these

estimates plotted as a function of distance around *Debbie* at landfall.

[Table 2 here]

[Figure 4 here]





### 4.3 Wave conditions

Wave observations during *Debbie* are shown in Figure 5. Maximum wave heights and wave conditions at the time of peak storm tide conditions are given in Table 3. The locations of the three wave buoys are shown in Figure 1.

[Figure 5 here]

[Table 3 here]

### 4.4 Wave runup and maximum water levels

Wave runup, $R_{2\%}$, and maximum water levels were estimated empirically using Eq. (2) for the three main impact sites to the south of *Debbie*; Midgeton Beach, Conway Beach and Wilson Beach (Figure 6, locations Figure 1). Given that storm tide was the major

component of total water levels, $R_{2\%}$ was calculated for wave conditions at the time of the highest storm tide (rather than maximum wave conditions). The maximum water level was defined as $R_{2\%}$ plus the maximum storm tide level, which was taken from the nearest gauge (4.4 m AHD, Laguna Quays).

At each site, three cross-shore profiles were taken through the pre-*Debbie* (July/August 2016) LiDAR DEM, extending from the landward edge of the foredune to as far seaward as possible. Because the LiDAR was not water-penetrating, the seaward extent of

profiles depended on the location of the waterline at the time of data capture. A single $R_{2\%}$ value was calculated for each site based on the mean slope (tan$\beta$) of the three profiles.

[Figure 6 here]

### 4.5 Field observations of maximum water levels

Estimates of maximum water levels were made by locating the most landward line of debris visible in the field. This can be difficult post-cyclone when the clean-up operation occurs very early after impact. RF/MQU took DGPS measurements of inundation markers at Seaforth, Laguna Quays and Midge Point, four days after *Debbie* (1 April 2017). DSITI undertook measurements at the same sites, in addition to Midgeton, Conway Beach, Wilson Beach and Hamilton Island, eight days after *Debbie* (5 April 2017). Results are summarised in Table 4. At sites where both groups captured data over the same area, the means compared well (a

difference of 0.02 m).

[Table 4 here]

### 4.6 Volumetric beach erosion

Digital Elevation Model (DEM) surfaces were created from pre- and post-*Debbie* LiDAR, at the three main study sites (Midgeton, Wilson and Conway Beach). The pre-*Debbie* DEM was subtracted from the post-*Debbie* DEM to derive a difference model at each location showing areas of erosion and accretion (Figure 7). The seaward extents of the difference models extended to 2 m AHD after analysis of the LiDAR data indicated this was the approximate position of the waterline during the pre-*Debbie* data capture.

[Figure 7 here]



Although we cannot be sure all morphological change in Figure 7 is attributable to *Debbie* (because the pre-storm data was collected eight months prior) it most probably did, especially on the upper beach area of interest, because a low wave energy regime with no storm conditions occurred during the eight-month period prior to *Debbie*.

Erosion volumes (m³) were calculated from these difference models for zones of interest at each site (Table 5, locations of erosion

zones in Figure 7). Indicative transport pathways were derived from a qualitative assessment of results.

[Table 5 here]

**4.7 Cross-shore beach profile change**

Pre- and post-storm cross-shore transects were taken through the LiDAR DEMs at Midgeton, Wilson and Conway Beach to

investigate the beach profile response to *Debbie* (Figure 8, locations Figure 7).

[Figure 8 here]

DGPS elevations were also surveyed along the same profile lines approximately five months after *Debbie* to investigate beach

recovery, if any. Only the lower profile of the beach is shown for the recovery surveys at Midgeton and Conway Beach. At Midgeton Beach, this was because heavy vegetation meant DGPS points could not be recorded for the foredune/reserve area. At Conway Beach, the upper portion is a rock revetment. No recovery profiles are shown for Wilson Beach because beach recharge had occurred in the interim.

The LiDAR-derived transects were used to calculate erosion and accretion volumes above 2 m AHD for each profile (Table 6).

The DGPS data were not used to derive cross-sectional area change, because of the high error (± 0.33 m) and because the surveyed lines were not exactly aligned to the LiDAR-derived transects. However, their inclusion clearly shows there has been very little (to no) beach recovery five months on from *Debbie*.

[Table 6 here]

**5 Discussion**

**5.1 Storm tide conditions**

The storm surge to the south of *Debbie*'s landfall location (Airlie Beach) was considerably larger than to the north, because of the clockwise rotation of low pressure systems in the Southern Hemisphere, east-facing coast and consequential onshore air flow to the south of the eye (Figure 3). The largest surge residual of 2.7 m and a maximum storm tide level of 4.4 m AHD was measured

at Laguna Quays, in Repulse Bay, approximately 40 km south of Airlie Beach (locations Figure 1). This was the highest water level recorded at this site, since the gauge was installed in November 1994. By comparison, the maximum surge reached at Bowen, 80 km to the north, was about two metres less because of the offshore-directed air flow (Table 2).

According to Hardy et al. (2004) and Haigh et al. (2014), the storm tide level at Laguna Quays equate to a recurrence of approximately 1,000 years. The storm tide during *Debbie* was similar to the level reached during TC *Yasi*, one of the most powerful

cyclones to have affected the Queensland coast, which made landfall further north near Cairns in 2011 with a storm tide level of 4.5 m AHD at Cardwell. The Cairns area has a smaller tidal range than Mackay and *Yasi* made landfall on a falling tide, so the



storm tide level comprised mostly surge (3.5 m), and thus was a much rarer event (Hardy et al. (2004) and Haigh et al. (2014) both estimate the *Yasi* storm tide to have a > 10,000-year recurrence).

It is important to note that these estimates are based on synthetic extensions of ~ 45 years of satellite derived TC track and intensity data (BoM, 2017) – which may not capture the longer-term variance of cyclone behaviour in the Coral Sea (South West Pacific).

For example, alternative methods that are not constrained by a short observational record (such as statistical- or paleo-based approaches) estimate the *Yasi* water level to have a lower 1,000 year recurrence (Nott and Jagger, 2013, and Lin and Emannuel, 2015). Extreme water levels as produced by *Debbie*, therefore, may be unprecedented with regards to recent observations, but not necessarily over the longer term.

Either side of Repulse Bay, water levels during *Debbie* were more than a metre lower, suggesting a more regionally representative

surge residual was 1.1 to 1.2 m. The higher water level recorded at Laguna Quays, and inundation observed at sites within Repulse Bay, suggest the surge was amplified in this area. Several geomorphological features may have been responsible for this, including; a shallow shoreface slope, concave shoreline planform and exposure to east and south-easterly winds with little sheltering by offshore islands (see Figure 1). Indeed, it may have been these features that contributed to the strong ebb current that first 'repulsed' Captain Cook in 1770, giving the bay its name.

South of landfall, wind setup was the biggest contributor to surge (accounting for 54 to 85 % of the total surge residual) because wind flow was onshore-directed. Conversely at Bowen, north of landfall, the inverse barometric (IB) effect was the largest contributor to surge (85 %). This was probably because the wind was offshore-directed at this location, causing a set-down in residual water levels (Figure 4). By comparison, McInnes and Hubbert (2003) found wind setup contributed ~ 90 % of the total surge during extra-tropical cyclone events (more than here), because the atmospheric pressure drop is not as significant.

Prior to *Debbie* crossing the coast, landfall was forecast to occur close to high spring tides – triggering evacuation advice for over 4,000 properties within the Whitsunday storm tide zone (IGEM, 2017). It eventuated that the slowing of *Debbie* on approach to the coast meant that landfall occurred approximately two hours after the spring high water, averting more widespread inundation. Maximum water levels occurred (in most locations) prior to landfall, coincident with the high tide, whereas the maximum surge residual occurred 1 – 2 hours after landfall, on the falling tide (Figure 2). This may be because of *Debbie's* south-west trajectory

(Figure 1) and slow forward speed, meaning the southern limb (and strongest winds on the south side) reached the coast sometime after the eye made landfall at Airlie Beach.

Another reason for the delay of the surge peak may be the role of tide/surge interactions. Analyses of tide gauges in the North Sea suggest that, in shallow water environments, tide/surge interactions can cause a phase shift in the total water level that means the maximum surge residual (which is the sum of the meteorological surge plus phase-shift effects) is more likely to occur on the

rising, and sometimes, falling tide, but almost never coincident with high tide (Horsburgh and Wilson, 2007). Tide/surge interactions are important determinants of surge maxima elsewhere in Australia (e.g. Bass Strait, Victoria - McInnes and Hubbert, 2003, and Broome, Western Australia - Haigh et al., 2014), but their role in modulating extreme water levels along the Mackay-Whitsunday coast requires further investigation.

**5.2 Wave conditions**

Wave conditions to the south of *Debbie* peaked 2 to 4 hours before landfall, broadly coinciding with high tide and peak water levels (Figure 5). The largest wave height recorded at Mackay (before buoy failure) was $H_{max}$ 8.7 m, but extrapolation suggests $H_{max}$ 10.7 m and $H_s$ 7.1 m may have occurred at the buoy location ($d$ = 35 m) (Table 3). De-shoaling these waves suggests that offshore (deepwater) conditions – representative of the edge of the GBR shelf area around 100 m depth - may have reached $H_{max0}$ 11.5 m and $H_{s0}$ 7.6 m. During this time, waves were coming from the east to north-east (onshore-directed) and travelling at around



$T_p$ 8 – 9 s, which conforms with the expected wind direction and fetch at this time. Wave heights have not exceeded 10 m at Mackay since the buoy record began in 1975 – meaning *Debbie* may have produced the most extreme wave conditions for this area over the past 40 years. Prior to this, TC *Dylan* produced $H_{max}$ 10.0 m in January 2014 and TC *Ului* $H_{max}$ 9.4 m in March 2010.

On *Debbie's* north side, waves were much smaller and still coming from the ENE (onshore) as *Debbie* approached the coast (Abbot

Point buoy, Figure 5), whereas the wind direction was blowing in a broadly offshore direction. This suggests the most energetic waves to the north of *Debbie,* until approximately 3 hours prior to landfall, were still being produced by ambient weather, rather than the approaching cyclone. At 10:00 on 28 March, *MWD* abruptly swung from 68 ° (ENE, onshore) to 257 ° (WSW, offshore) as the offshore winds strengthened as *Debbie* moved closer to the coast, becoming the main generation source for local waves. At the same time, $T_p$ dropped from 9.1 to 4.4 s, again indicating the dominance of a local, offshore wind forcing.

Because of *Debbie's* slow forward-moving speed, damaging storm wave conditions were sustained for an unusually long period of time (approximately 220 hours), beginning three days before *Debbie* made landfall to two days after. The long duration meant a considerable amount of wave power was distributed along the coast (cumulative storm wave power, $P_c$, at Mackay buoy was ~ 7.4 mWh m, with a peak $P_0$ of 0.24 mWh m). By comparison, the duration and cumulative wave power for the previous two landfalling cyclones in this area were 91 hours and 3.8 mWh m (TC *Dylan*), and 174 hours and 5.2 mWh m (TC *Ului*). For extra-

tropical cyclones impacting the Queensland and New South Wales coasts, indicative duration and $P_c$ values are around 90 hours and 3.0 mWh m (data from Shand et al., 2011 and Splinter et al., 2014, using the same storm identification method as here). *Debbie*, therefore, can be regarded a significant wave power event not only locally, but for the whole east coast of Australia.

**5.3 Coastal inundation and wave runup**

Maximum limits of coastal inundation were estimated by summing the local measured storm tide with an empirically-derived

estimate of wave runup, $R_{2\%}$, using the approach of Stockdon et al. (2006). These estimates were then compared with observations made in the field directly after *Debbie*.

At Midgeton and Conway Beach, $R_{2\%}$ and the maximum water level were estimated at 0.93 m and 5.33 m AHD (Figure 6). This water level would have breached the dune at profiles MP1 and CB1 (southern end of Midgeton and Conway Beach) where the dune crest is lower. The difference plots in Figure 7 also indicate dune breaching and overwash occurred in these areas. The

estimate of ~ 5.3 m AHD at Conway Beach broadly agrees with field evidence at this location (~ 5.1 m AHD, Table 4).

At Wilson Beach, $R_{2\%}$ and the maximum water level were estimated higher, at 1.60 m and 6.00 m AHD. This is because the upper beach at Wilson Beach is steeper (tan$\beta$ 0.13) than Midgeton and Conway Beach (0.02 – 0.03), increasing the estimation of wave runup in Eq. (2). This water level would have breached the dune at every profile along Wilson Beach and caused significant coastal flooding to residential areas. Field surveys of debris lines and photographic evidence (Figure 9) show this did occur at this location.

However, post-*Debbie* measurements indicate the maximum water level caused by coastal flooding was lower, between 5.1 and 5.2 m AHD, broadly similar to other open-coast sites (Table 4). This water level was still able to breach the dune crest across the whole of the beach frontage. The empirical over-estimate at Wilson Beach may be due to the large difference in the slope of the upper (tan$\beta$ 0.13) and lower (tan$\beta$ 0.01) beach, and that Eq. (2) assumes incident waves (no refraction) which is most likely not the case here. Stockdon et al. (2006) note that use of Eq. (2) for reflective beach types with variable cross-shore slopes can indeed lead

to large errors in the estimation of runup. In addition, the storm tide peak at Laguna Quays may have reduced by the time it reached Wilson Beach due to the shallow bathymetry at the Proserpine River entrance and attenuation of the tidal wave.

[Figure 9 here]





At Midgeton Beach, field measurements suggest inundation was lower than at other sites, the empirical estimate, and the local storm tide measurement, reaching only 4.3 m AHD (Table 4). This is likely to be because Midgeton town sits at a lower elevation than the fronting dunes. For the water to have reached the locations at which debris lines were measured, it must have first breached the dune crest, which is between 5 and 6 m AHD.

At Laguna Quays, the storm tide displaced pontoons from their moorings and deposited them on adjacent grass land, providing a useful inundation marker, at around 4.6 m AHD (Figure 9). Laguna Quays marina is sheltered from most wave effects; therefore, the total water level only includes the storm tide with perhaps some small additional wave motion. For this reason, the field estimates are close to the maximum storm tide level recorded at the gauge (4.4 m AHD).

At Seaforth, maximum water levels were around 5.2 to 5.3 m AHD along the central frontage, and 3.8 to 4.1 m AHD at the more

sheltered northern and southern ends of the beach. The highest inundation occurred at Hamilton Island, where the north-east facing beach experienced water levels up to 5.9 m AHD. Incidentally, Hamilton Island was the location of the highest recorded wind gust during *Debbie*, of 260 km h$^{-1}$ (uncorrected for topography and around 240 km h$^{-1}$ after correction to standard conditions, Boughton et al., 2017).

The storm tide contributed between 70 to 98 % of total water levels at the mainland, open-coast sites (Conway Beach, Wilson

Beach, Midge Point, Seaforth), with a mean of 84 % (Table 4). By comparison, previous work suggests the storm tide contributes between 65 to 75 % of the total inundation of extreme paleo-cyclones along the North Queensland coast (Nott et al., 2009; Forsyth et al., 2010). The higher contribution of storm tide during *Debbie* reflects the macro-tidal range of the Mackay area, which significantly reduces further north along the coast. The large tide makes it difficult to make accurate TC storm tide forecasts, because a few hours error can translate to hundreds of metres of difference in the cross-shore position of the waterline.

Wave effects were responsible for 2 to 30 % of total water levels, depending on coastal exposure, with a mean of 16 %. The empirical estimate of Stockdon et al. (2006) did reasonably well in replicating the relative contribution of waves to total water levels. If a regional upper beach slope of 0.02 – 0.03 is used (as per Short (2006) and measured at Midgeton and Conway Beach), and the mid-shelf wave record is de-shoaled to deep-water values, then wave effects are estimated at approximately 17 % of the total inundation. This appears to capture the mean, regional contribution of waves to total water levels, assuming sites receive

incident wave energy and do not deviate far from the regional mean slope.

## 5.4 Patterns of coastal erosion

At Wilson Beach, the LiDAR surveys suggest *Debbie* caused erosion to the whole of the upper high tide beach (displacing a volume of approximately 4,500 m$^3$, Table 5), with some material deposited immediately shoreward of the beach crest (Figure 7 A). This suggests maximum water levels breached the barrier dune across the whole frontage at Wilson Beach.

At Conway Beach, a rock revetment protects the township at the western end of the bay (denoted in red, Figure 7 B), while the eastern end is a natural system with vegetated dunes and a small creek behind the beach. The undefended section showed similar behaviour to Wilson Beach during *Debbie*; erosion to the upper beach (approximately 960 m$^3$) and overwash (deposition) of some of this material landward of the dune crest. In front of the revetment, however, the beach response differed. Figure 7 B indicates the upper beach erosion (approximately 2,500 m$^3$) was deposited seaward at the toe of the upper beach, rather than transported

landwards.

Overall, the revetment at Conway Beach appears to have prevented significant overtopping. Figure 7 B does, however, suggest there was some overwash of material at the west end between profiles CB1 and CB2, which concurs with the empirical estimates in section 5.3 that maximum water levels around CB1 breached the revetment crest (Figure 6 B). In addition, there is an area of large erosion (beach lowering by up to 1.5 m) directly to the east of the termination of the revetment - perhaps an 'end effect' of





the rock wall. End-effects occur when erosion is focussed on areas directly down-drift of a coastal defence, when the structure protrudes seaward of the natural beach crest (as it does at Conway Beach).

At Midgeton Beach, erosion was focussed on the central/north section of the beach, in front of the township of Midgeton (approximately 12,500 m³, Table 4). The LiDAR data suggests most of the eroded material accumulated at the toe of the high-tide
beach in a depositional lobe about 300 m in length. This feature probably extended further seaward than was captured in the DEMs, and may have resulted from a rip-cell circulation that was set up here, transporting eroded material alongshore and then offshore in front of the central beach.

There is little evidence of much landward transport (overwash) of sand at Midgeton. Likewise, the dunes did not experience any notable roll-over as seen at Wilson Beach (Figure 8 A – C). Instead, the beach responded similarly to the revetted section at Conway
Beach, where eroded sand was transported seaward. This may be related to the reinforced stability of the coastal foredunes at Midgeton; a community-led initiative had ensured they were well-vegetated and in most part underlain by a geotextile mat prior to *Debbie* (Zavadil et al., 2017).

Another commonality between Conway and Midgeton Beach was the major erosion that occurred near tidal creeks. At Conway Beach, storm water transported seaward down the creek caused large, localised erosion at the mouth (~ 3,700 m³ - more than the
rest of the entire frontage), and was deposited at the toe of the upper beach. The difference plot is cut off at + 2 m AHD, but suggests a 200 m-wide ebb tidal delta formed in this area.

Similarly, at Midgeton Beach, significant erosion occurred at the two creek entrances, with the southern creek eroding an almost identical volume to the Conway Beach creek (~3,700 m³). Sand from both the northern and southern creeks looks to have washed out in a south-westerly direction and been deposited offshore.

**5.5 Storm erosion demand and offshore transport**

The storm erosion demand during *Debbie* was estimated from the beach profiles extracted from the pre/post storm LiDAR DEMs. Storm demand refers to the erosion that occurs to the upper beach during a storm event (typically above 0 m AHD in Australia). It is an important parameter used for the design of coastal setback. Here, the storm demand can only be calculated above 2 m AHD because this was the limit of the LiDAR data. However, beach profiles show that all erosion occurred well above 2 m AHD (Figure
8); therefore, volumes should be comparable to those calculated to 0 m AHD in other studies.

The average storm demand at Midgeton Beach was 15.6 m³ m, and at Conway and Wilson Beach it was roughly half this (7.3 and 9.2 m³ m, respectively) (Table 6). The lower volumes at Conway and Wilson Beach may be because these locations are afforded some protection from easterly waves by Cape Conway, whereas Midgeton Beach is open to both easterly and south-easterly wave attack (Figure 1).

These volumes are an order of magnitude smaller than those of wave-dominated coasts (e.g. 150 – 250 m³ m is a typical range for New South Wales and Southeast Queensland) but comparable with modelling studies of the Mackay coast, which suggest the 100-year ARI erosion volume is between 1 and 16 m³ m, depending on aspect (Mariani et al., 2012). To our knowledge, this study represents the only set of observations of TC storm erosion demand for north Queensland.

Most profiles saw a net loss of sand above 2 m AHD (average 78 % loss, Table 6), suggesting that the eroded volume was
accumulated below (seaward of) this level. The exceptions were CB2 and CB3 at Conway Beach which experienced a net gain above 2 m AHD, because the depositional lobe that accumulated at the toe of the upper beach extended into the surveyed portion of the profiles. This suggests that at all study sites, sand transport was predominately offshore. At Midgeton and Conway Beach, this may have been in part due to the stabilisation of the coastal frontage. Even at Wilson Beach, however, where the coastal





foredune was not stabilised, results indicate that most sediment was transported seaward of the high-tide beach (beyond the limits of our data capture), with only a small portion of the eroded volume overwashed landward.

Our results suggest that while overwash and landward transport clearly occurred during *Debbie*, this only accounted for a small portion of the total eroded volume of the upper beach, and the majority was transported offshore and placed around the toe of the high-tide beach. This may have been because at most sites, coastal water levels did not fully breach the foredune, inducing offshore transport. Similarly, Sallenger et al. (2006) found that the net direction of sediment transport during Hurricane conditions along the US East Coast is a function of the water level relative to the foredune height (i.e. overwash occurs when this ratio is positive, else there is offshore transport). While this may be true at the peak water level, as the tide recedes and storm wave conditions continue, erosion is focussed lower down the profile and sediment may be transported offshore. The result can be a mix of on- and off-shore transport even at sites inundated at the peak water level (as was the case at Wilson Beach).

## 5.6 Beach recovery and implications for coastal management

Beach profile measurements taken five months after *Debbie* show very little, if any, beach recovery occurred over this period. While the quality of the DGPS measurements was not sufficient to derive volumes, their inclusion in Figure 8 shows the profiles are very similar to those captured directly after *Debbie*.

This can be attributed to the wave climate of the region. Tropical cyclones often impact coastal areas that are equilibrated with a low-energy, tide-dominated regime because of their latitude and small regional wind-wave climate. This is particularly the case along the North Queensland coast, where a significant amount of wave energy is dissipated over the Great Barrier Reef (Gallop et al., 2014). This is exemplified in the wave buoy record pre- and post-*Debbie*; prior to *Debbie*, storm wave conditions ($H_s \geq 2$ m) had not been exceeded since the start of January (3 months prior). Likewise, after *Debbie*, it took over three months – to mid-June – to accumulate the same amount of wave power as was exerted over nine days during *Debbie*. It was at this point (June 2017) that the beach recovery surveys were undertaken.

The large energy difference that exists between modal and cyclonic wave conditions in this region means TC erosion impacts have a permanent impact on the landscape and require a management response to restore beach amenity and access. In some geomorphic settings natural rebuilding of the frontal dune will occur over time, however this may take a much longer period than between successive cyclones. For example, photographic evidence 12 months on from *Debbie* suggests some minimal sand accretion had begun to occur at Midgeton Beach (Zavadil *pers. comms.*)

Where natural recovery processes occur, coastal management actions should be designed to encourage and accelerate natural rebuilding of the beach. For example, the shoreline at Midgeton Beach has eroded and recovered many time over recent decades (Zavadil et al., 2017) and the underlying tendency for many beach ridge systems in North Queensland is progradation. Natural recovery processes become important for coastal management particularly when engaging local communities about erosion management options.

## 5.7 Damage to buildings

Although *Debbie's* surge did not coincide with the highest tide, some buildings at Wilson Beach and Hamilton Island suffered varying degrees of damage from storm tide and waves. The most severe structural damage was observed at Wilson Beach. Field observations indicated three processes were responsible; 1) overland flow causing scour around footings, foundations, sub-floor structures and piles; 2) lateral forces exerted by waves on external cladding elements such as doors and windows, and; 3) direct inundation causing damage to internal wall linings, floors and building contents.



Fast-flowing water associated with in-flowing (landward) and out-flowing (seaward) storm tide inundation can cause localised scour around a building and its foundations. In-flow tends to be more uniform across an area, but is exacerbated by wave action. Post-*Debbie* surveys suggested that the out-flow was not powerful enough to develop overland ebb channels, but that both in-flow and out-flow combined caused some scouring of building footings with subsequent settlement (Figure 10 D).

[Figure 10 here]

In areas affected by tidal creeks, the in/out flow pattern was further complicated. At Wilson Beach, the storm tide entered from two directions; the south (beach) side, where buildings were directly exposed to wave action, and the north (creek) side, which was
responsible for most inundation of the community (Figure 10 A). Wave action on beach-side properties caused damage to external cladding elements and broke windows and doors on some houses. Wave action also caused substantial drag forces on the substructure of buildings with suspended floors. The window and cladding damage shown in Figure 10 D occurred as waves broke against the wall of the house after it had been knocked off its piles. Inundation by storm tide and waves caused further internal damage to fittings, linings, electrical outlets, floor coverings and building contents (Figure 10 B), and the receding water left a
layer of mud on wall linings and floors (Figure 10 C).

At present, building damage curves that relate cyclone intensity to structural damage treat wind and water damage separately. Field inspections post-*Debbie* suggest that the upward wind action exerted on roofs combined with the upward wave and storm tide action on the underside of floors, in some instances, combined to wash buildings off their stumps. This highlights the need to develop TC building damage relationships that account for the combined impacts of water and wind, rather than treating them
separately.

**5.8 A shared information platform for coastal extremes**

Aside from the technical findings of this research, this paper provides a case study of data sharing across industry, government and academia in a natural hazard risk context. The research was facilitated through an information-sharing event (the TC *Debbie* forum, organised by the Bureau of Meteorology and hosted by the Queensland Fire and Emergency Service) and is built on the principles
of openness and collaboration. Each group collected data independently, according to their own time and cost constraints. Not all data was collected with coastal hazard monitoring as the primary purpose (e.g. Fugro Roames LiDAR). The value added by unifying this information came at minimal additional cost, but was essential to understanding the relationships between the extreme hydrodynamic forcing and coastal impacts.

At a high level, the economic value of data sharing is well recognised (e.g. Deloitte, 2014, World Bank, 2014), but in Australia
cross-sector data exchange during natural disasters is lacking (Gissing, 2017). Data sharing can lead to direct and (mostly) indirect economic benefits such as efficiency gains, a reduction in competitive bias, better planning and prediction by regulatory agencies, improved price signalling by insurance, and facilitates research and innovation. However, reluctance to share information, a lack of co-ordination and standardisation and high costs of data collection often restrict data sharing and encourage the continuance of a piece-meal approach.

In Australia, it is estimated that the centralisation of key natural peril data through the development of open data platforms could save the economy over $2 billion over the period to 2050 (Deloitte, 2014). Because of rising coastal population density (both globally and in Australia), coastal extremes are likely to become particularly costly events. We believe a shared information platform is needed for coastal extremes in Australia, to provide a repository of observational coastal data for historical extremes. In this way, all relevant data and metadata is indexed on an event-by-event basis in a single location. This may include information



on the meteorology (winds, air pressure), hydrodynamics (waves, water levels), coastal response (erosion, inundation) and structural (water) damage aspects of individual events that have impacted the Australian coast. At present, a large number of organisations collect this type of information (and most is, in principle, publicly available) but it is often siloed within a discipline or sector. The centralisation of this data would simplify access and increase its collective value, as has been demonstrated in this

paper.

## 6 Conclusions

Tropical Cyclone (TC) *Debbie* was potentially one of the most powerful wave- and surge-producing cyclones to have made landfall on the northeast coast of Australia since monitoring began. Despite localised impacts, *Debbie* was a near-miss in terms of widespread coastal flooding. If landfall had occurred two hours earlier (i.e. if the surge had coincided with high tide), then the

maximum storm tide may have been ~ 18 % higher. Total water levels would likely have exceeded 6 m AHD on the southern side, which is higher than most dune crest elevations. It eventuated that *Debbie* slowed on approach to the coast meaning landfall occurred on the falling tide, averting widespread inundation.

The maximum recorded storm tide was 4.4 m AHD, and the maximum recorded surge was 2.7 m (Laguna Quays). These were the highest levels recorded at this location (since the gauge was installed in 1994), with previous work suggesting a storm tide

recurrence of approximately 1,000 years. These maxima are likely the result of surge amplification within Repulse Bay with sites north and south recording levels over a metre lower. The considerable variation in water levels over relatively small spatial scales (~ 20 km between gauges) highlights the need for suitably high resolution coastal models and data that can capture the effects of local-scale geomorphology on the hydrodynamics.

Waves, water levels and coastal impacts were considerably larger south of the landfall site, because of the clockwise rotation of

low pressure systems in the Southern Hemisphere, east-facing coast and onshore air flow to the south of the eye. An average coastal water level (storm tide plus waves) for the region Airlie Beach to Mackay was 5.2 m AHD, ranging between 3.8 m to 5.9 m AHD depending on exposure. Total inundation varied dramatically close to tidal creeks. At Wilson Beach, flooding was caused by the storm tide entering the creek and inundating the town from the behind, in addition to coastal flooding. The largest beach erosion volumes were also found near to tidal creeks at Conway Beach and Midgeton Beach.

Results suggest the storm tide contributed ~ 84 % of total water levels, which is higher than previous estimates for tropical cyclones in Australia (65 – 75 %), but may reflect the locally large tidal range. Wave runup was responsible for ~ 16 % of total water levels, varying between 2 and 30 % at sites depending on coastal exposure. Wind setup contributed between 54 and 85 % of the total surge residual south of landfall, but probably contributed to a set-down in water levels north of landfall.

*Debbie* may have produced the most extreme wave conditions in this area for at least the past 40 years, with an (extrapolated)

maximum wave height exceeding 11 m. The slow forward-speed led to an unusually long duration of storm waves and large cumulative wave power (7.4 mWh m), that began three days before landfall and continued for another eight days. This caused significant beach erosion in areas exposed to easterly and south-easterly waves, with storm demand volumes between 6 and 19 m³ m. Most of the eroded material was placed near the toe of the high-tide beach. While landward transport clearly occurred, this only accounted for a small portion (~ 25 %) of the total eroded volume of the upper beach. In most places, coastal barriers were not

breached and this may have been the cause for the lack of overwash deposits.

The large energy difference that exists between modal and cyclonic wave conditions in this region (and many other TC-affected global coastlines) means TC erosion impacts are often a permanent feature on the landscape within planning timeframes and require



a management response to restore beach amenity and access. Surveys undertaken five months on from *Debbie* indicated very little beach recovery had occurred. Photographic evidence 12 months on also indicated very small-scale geomorphic change.

To our knowledge, this study represents the only observational analysis of hydrodynamic drivers and concurrent coastal impacts for a tropical cyclone in Australia, and the only set of observations of TC storm erosion demand in north Queensland. The data

were collected on a largely ad-hoc basis on a limited budget. We advocate for a more formalised collaborative approach to collecting and archiving TC coastal impact data in Australia, as simple as a publicly available, standardised data depository, to help improve our understanding and prediction of TC-related coastal hazards in the future.

**Author contribution**

TM conceived the idea for the paper and prepared the manuscript. TM prepared the manuscript with contributions from all co-

authors, in particular DM who provided significant input on hydrodynamic forcing. Coastal impact surveys were carried out and analysed by TM, JS, SBL, JM, GB and EZ. IDG contributed to the analysis of field observations and coastal process implications.

**Competing interests**

The authors declare that they have no conflict of interest.

**Data availability**

The storm tide and wave buoy data used in this study (except for Abbot Point wave buoy) are maintained by the Coastal Impacts Unit, Department of Environment Science, (DES), Queensland Government and is available at https://data.qld.gov.au/dataset. The Abbot Point wave buoy is funded by the North Queensland Bulk Ports Corporation (NQBP) and data from this buoy were made available by NQBP for this study. All coastal field inspection data collected by DSITI is available in DSITI (2017). Inquiries regarding Risk Frontiers/Macquarie University field data, Fugro Roames LiDAR data or structural damage survey information

collected by James Cook University should be directed to the authors.

**Acknowledgements**

All coastal impact surveys were funded independently by each contributing organisation. The Cyclone Testing Station investigation was funded by the Queensland Department of Housing and Public Works, Australian Building Codes Board, and other CTS sponsors and benefactors. The authors would like to thank Bureau of Meteorology (BoM) Queensland State Manager Bruce Gunn

for organising the Tropical Cyclone *Debbie* Forum, hosted by Queensland Fire and Emergency Services (QFES) at the QFES Emergency Services Complex, Kendron, Brisbane, Australia, in June 2017, where the initial concept for this paper was conceived. We also thank Barry Hanstrum, previously BoM Regional Director for New South Wales, for facilitating this event and for providing comments on the manuscript. Thanks also to Ultimate Positioning Group (UPG) for providing access to the Trimble VRS Now base station network for post-processing the Risk Frontiers/Macquarie University survey data.



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

prepared by Alluvium and JBP for Mackay Regional Council, pp 55, October, 2017.



**Figures**

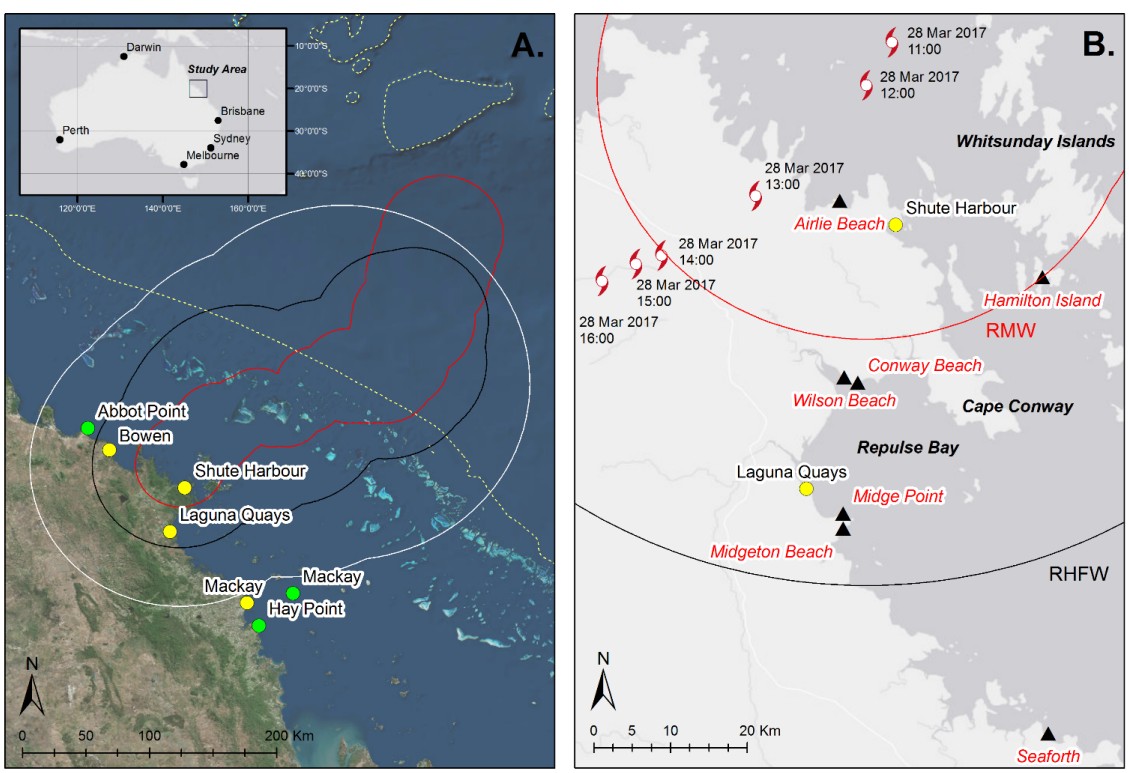

**Figure 1:** (A) Map of study area, with regional inset, showing *Debbie* best track between 00:00 27 March 2017 Australian Eastern Standard Time (AEST, UTC + 10 hrs) to landfall (12:40 28 March) with radius of maximum winds (RWM, red), radius of hurricane-force winds
5   (RHFW (≥ 33 m/s), black) and radius of storm-force winds (RSFW (≥ 25 m/s), white) (BoM, 2017). Locations of storm tide gauges (yellow dots) and wave buoys (green dots) used in this study are also shown. The 100-m isobath, which approximates to the edge of the Great Barrier Reef shelf, is also shown (dotted yellow line, from Bearman, 2010). (B) also shows locations of study sites (black triangles) referred to in this study.




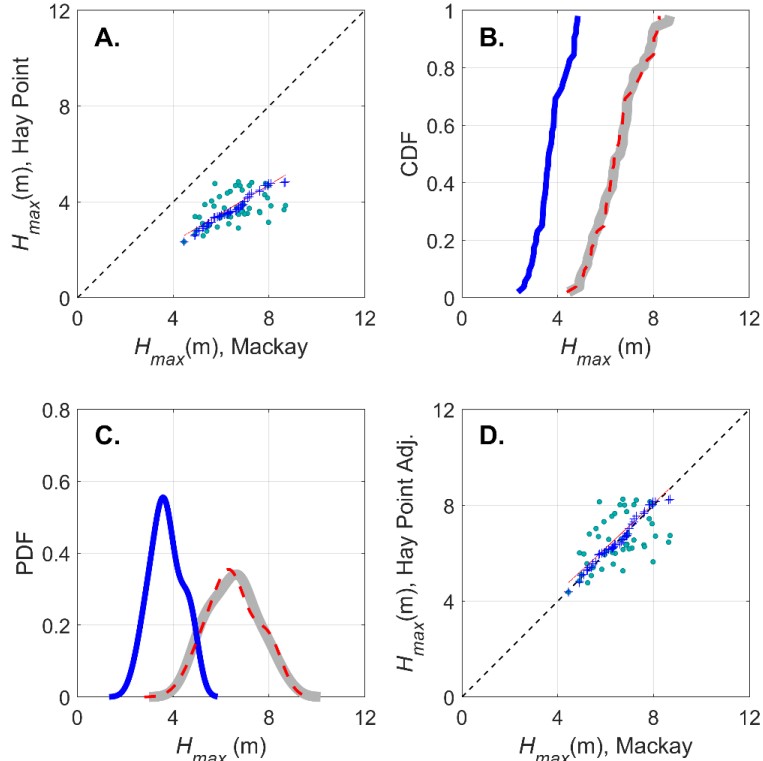

**Figure 2:** Example of the CDF mapping approach used to extrapolate an estimate of $H_{max}$ at the Mackay wave buoy, using data from the Hay Point buoy. (A) shows the raw observations (green) and quantiles (blue) of $H_{max}$ at both sites during the overlapping period; (B) shows the cumulative distribution of $H_{max}$ at Hay Point (observed, blue line), at Mackay (observed, grey line), and Hay Point (CDF-mapped to Mackay, red dotted line); (C) shows the same for the probability density of $H_{max}$, and (D) shows the CDF-adjusted Hay Point data against the original Mackay data, demonstrating its improvement as an predictor for $H_{max}$ at Mackay, compared to (A).





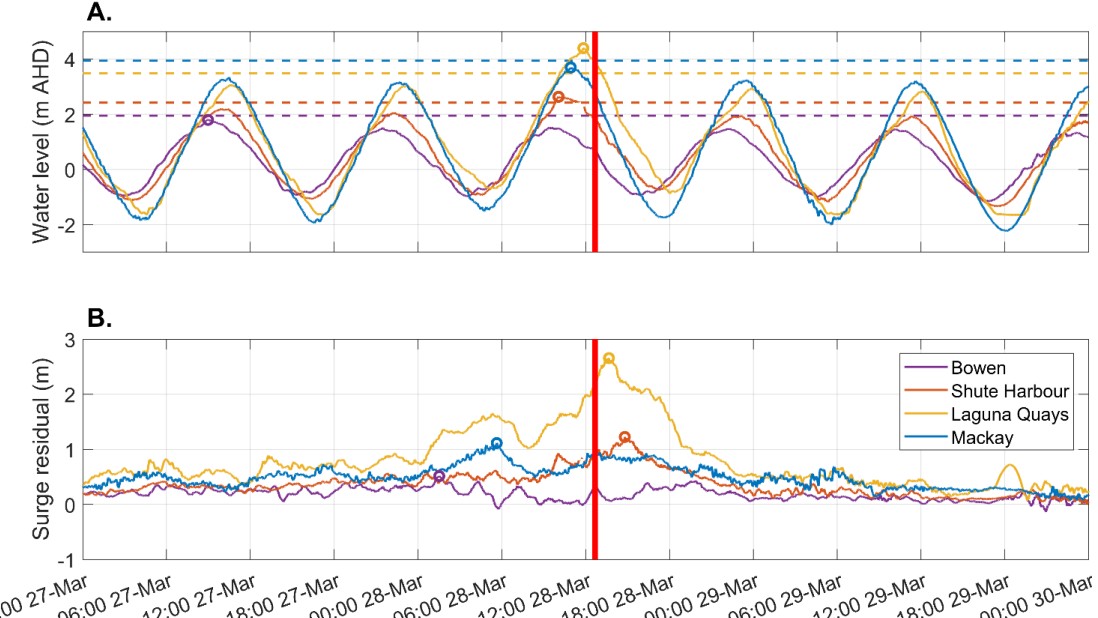

**Figure 3:** Water levels (A) and storm surge residuals (B) observed at (north to south) Bowen, Shute Harbour, Laguna Quays and Mackay storm tide gauges during TC *Debbie*. Highest Astronomical Tide (HAT, dotted horizontal lines), times of maximum water level (circles in A) and times of maximum surge residuals (circles in B) are also shown. Red vertical line indicates approximate time of landfall at Airlie Beach. Time is in Australian Eastern Standard Time (AEST, UTC + 10 hrs). Track of *Debbie* during this period (up to landfall) can be seen in Figure 1.





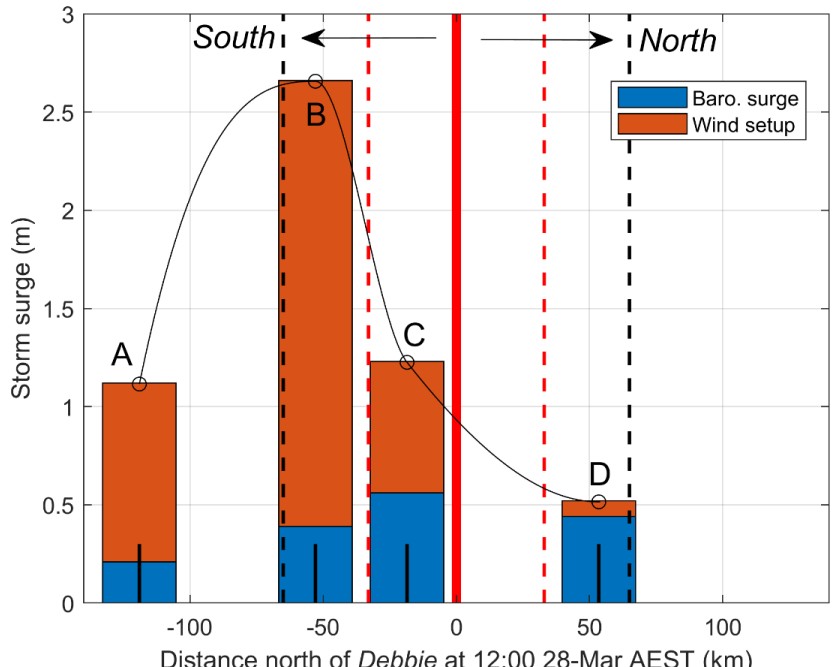

**Figure 4:** Maximum storm surge and components (barometric surge, blue, and wind setup, orange) plotted as a function of distance from *Debbie* landfall at 12:00 on 28 March 2017 (solid red line), where A = Mackay, B = Laguna Quays, C = Shute Harbour and D = Bowen storm tide gauge stations (black lines show centroids of each). Radius of maximum winds (RMW, red dotted lines), radius of hurricane-force winds (RHFW, black dotted lines) are also shown. The maximum surge north and south of *Debbie's* eye (black curve and circles) demonstrates the asymmetry in the surge, skewed to the south of the cyclone centre.



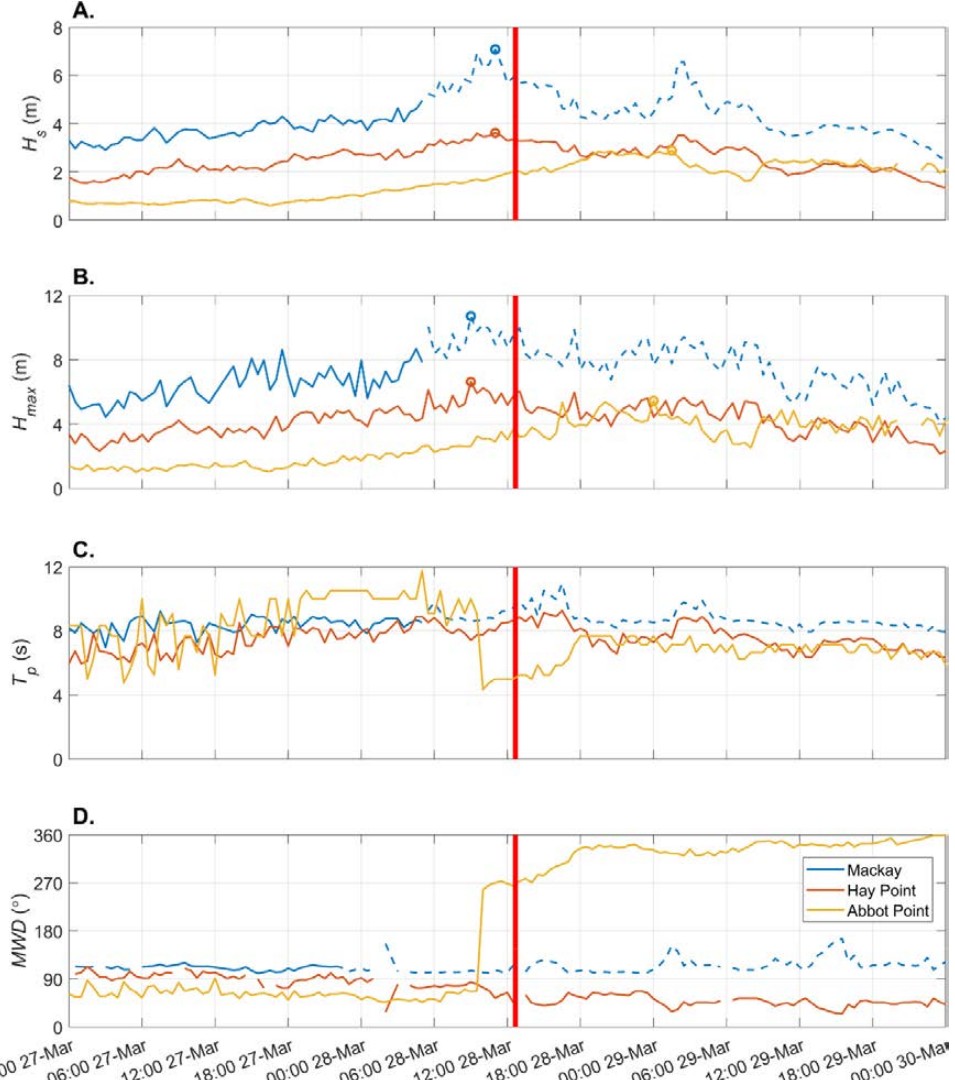

**Figure 5:** Wave observations during *Debbie* at (north to south) Abbot Point, Mackay and Hay Point wave buoys. Parameters shown are (A) significant wave height, $H_s$, (B) maximum wave height, $H_{max}$, (C) peak wave period, $T_p$ and (D) mean wave direction, *MWD*. Red vertical line indicates approximate time of landfall at Airlie Beach. Dotted blue line indicates the extrapolated estimate of wave conditions at Mackay, after the failure of the wave buoy at 05:30 on 28 March. Maximum $H_s$ and $H_{max}$ at each buoy are also shown in (A) and (B), respectively (circles).





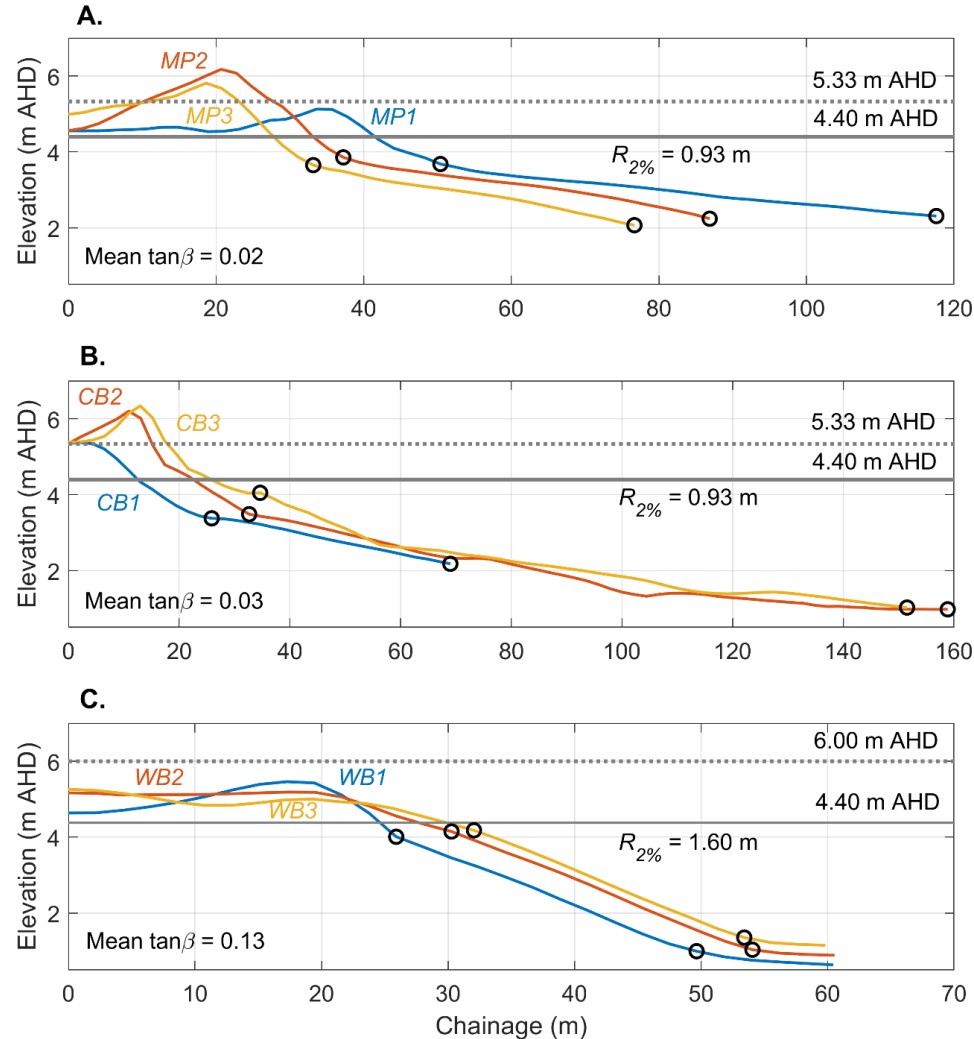

**Figure 6:** Pre-*Debbie* beach profiles (July-August 2016) at (A) Midgeton Beach, (B) Conway Beach and (C) Wilson Beach (profile locations in Figure 7). Maximum storm tide elevation was taken from the nearest gauge (Laguna Quays) (grey solid line), and maximum water elevation (grey dashed line) was defined as $R_{2\%}$ + storm tide elevation. $R_{2\%}$ corresponds to the expected level of exceedance of 2% of the wave runups during wave conditions at the time of maximum storm tide elevation ($H_{s0}$ 5 m $L_0$ 131 m, at 12:00 28 Mar). $R_{2\%}$ was calculated for each location based on the mean slope (tan$\beta$) of the profiles. The locations at which tan$\beta$ was calculated for each profile are shown (base of dune to toe of upper beach, black circles).



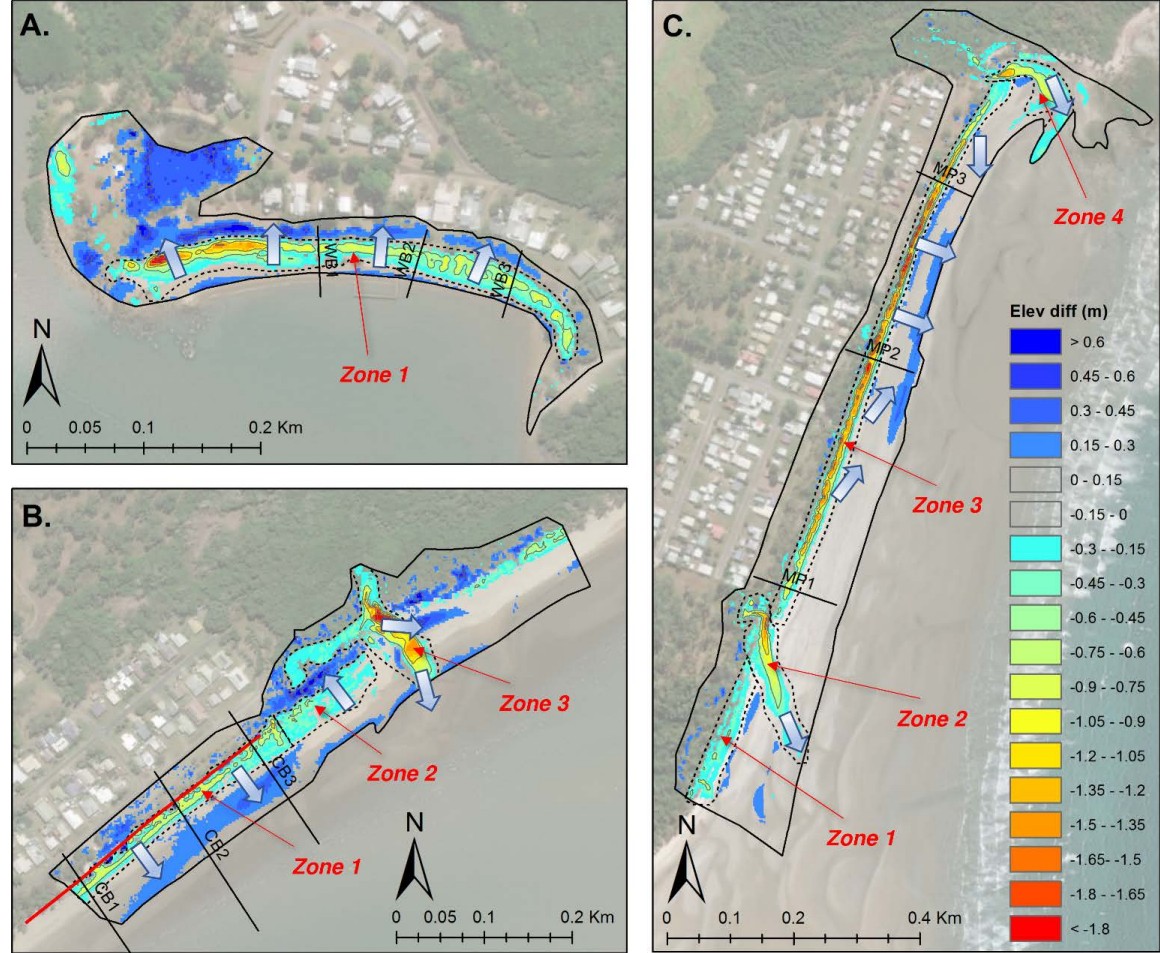

**Figure 7:** Coastal erosion, deposition and inferred transport pathways during *Debbie* at (A) Wilson Beach, (B) Conway Beach and (C) Midgeton Beach. Elevation difference plots show change in metres (pre-*Debbie* minus post-*Debbie*), with ± 0.15 m (reported vertical error of LiDAR) not shown. Contours at 1.5, 1.0, 0.5, -0.5, -1, -1.5 and -2.0 are overlaid (grey lines). The approximate position of the rock revetment at Conway Beach is shown in (B) (red line). Also shown are locations of cross-shore beach profiles at each site, extents of Fugro Roames LiDAR data used (black box), and erosion zones referred to in the text (dashed lines).



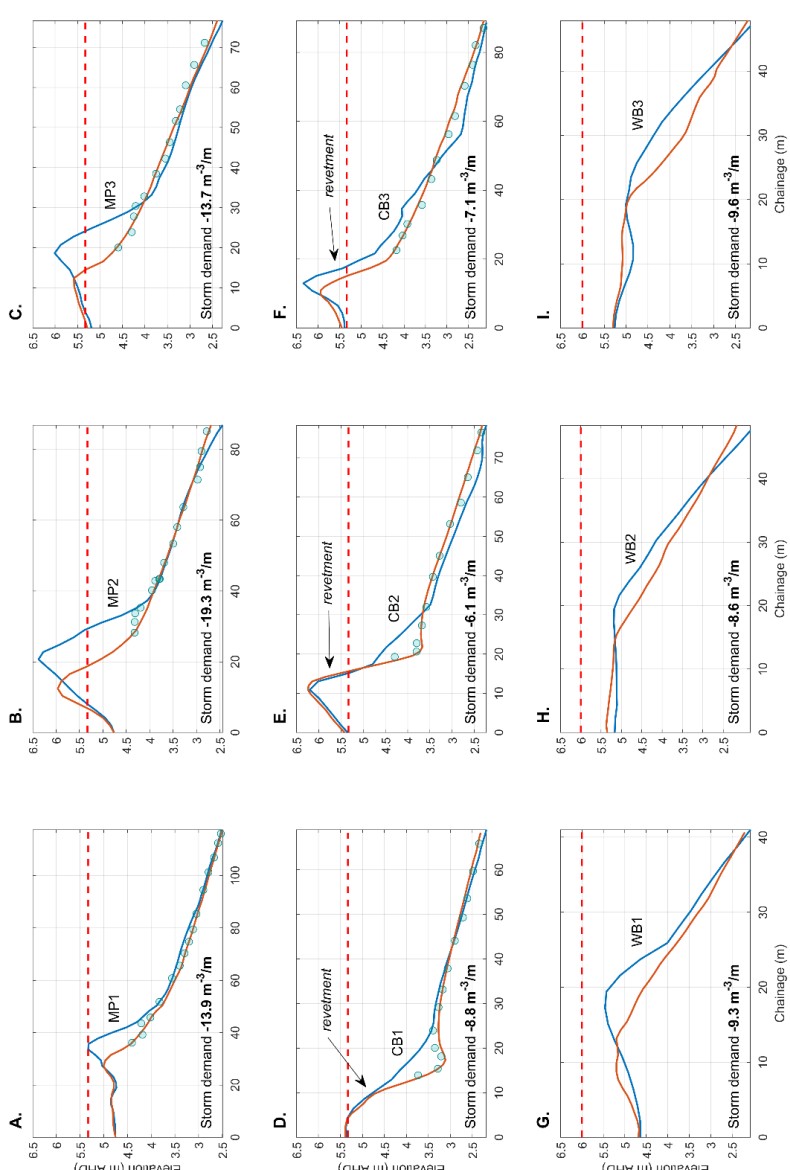

**Figure 8:** Cross-shore beach profile change at Midgeton Beach (A – C), Conway Beach (D – F) and Wilson Beach (G – I). Blue (orange) lines denote pre- (post-) storm profiles taken through the LiDAR DEMs (locations Figure 9). Green circles show alongshore-averaged DGPS elevations along the same profile lines ~ 5 months after *Debbie*. Only the lower 'recovery' profile is shown at Midgeton Beach because heavy vegetation hindered DGPS data collection. Only the lower 'recovery' profile is shown at Conway Beach because the upper portion is a revetment. No 'recovery' profiles are shown for Wilson Beach because beach recharge occurred in the interim. Also shown is the estimate of maximum water elevation from Figure 6 and storm demand. (F) indicates the revetment has moved during *Debbie* along profile CB3, consistent with observations (DSITI, 2017). The storm demand calculation for CB3 does not include the revetment change.




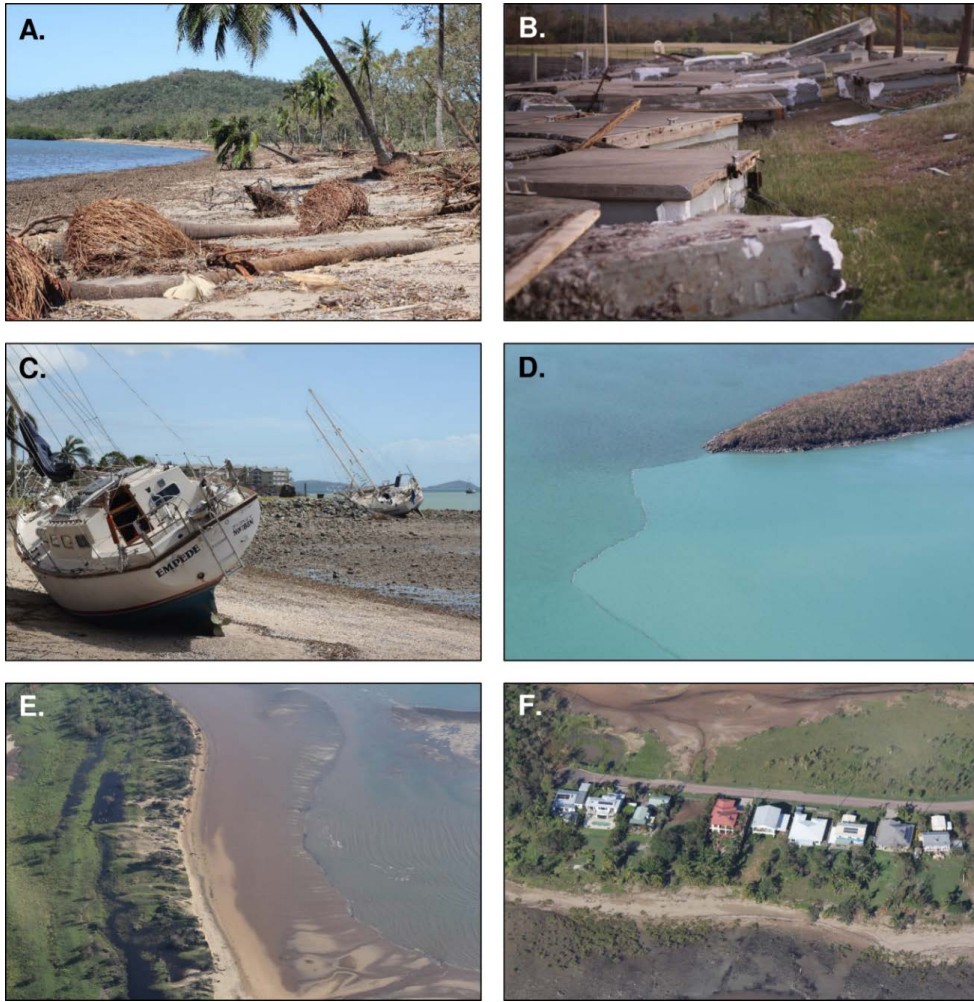

**Figure 9:** A selection of post-*Debbie* ground and aerial images. Erosion of the beach and dunes at Seaforth (A); concrete-clad pontoon displaced on adjacent grassland, providing an inundation marker, at Laguna Quays Marina (B); yachts stranded on Airlie Beach (C); brackish river runoff meeting seawater causing flocculation around Hamilton Island (D); an example of natural barrier overwash and pooling (E) and the contrasting exposure of some residential property to coastal processes (F), when development occurs within these natural overwash zones. Authors' own images.



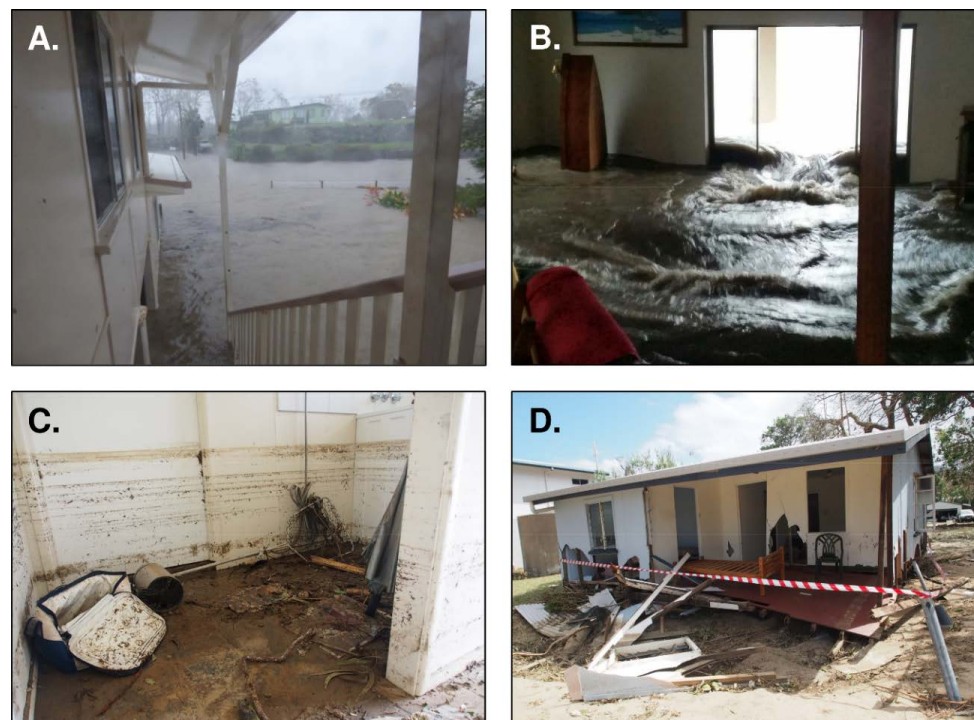

**Figure 10:** Flooding (A), water ingress (B), interior (C) and exterior (D) damage to property at Wilson Beach. (C) and (D) are authors' own images; (A) and (B) were taken by residents at Wilson Beach, reproduced with their permission for this paper.





**Tables**

**Table 1.** Storm tide gauges and wave buoys used in this study (north to south). AHD conversion from local LAT is given if site is a storm tide gauge, else mooring depth of water is given for wave buoys.

| Site name | Location (Lat, Lon) | Type | Start of record | AHD from LAT (m) *or* Depth (m) |
|---|---|---|---|---|
| Abbot Point | -19.87°, 148.10° | Wave buoy | January 2012 | 14 |
| Bowen | -20.02°, 148.25° | Storm tide gauge | March 1975 | -1.78 |
| Shute Harbour | -20.29°, 148.78° | Storm tide gauge | July 1976 | - 1.91 |
| Laguna Quays | -20.60°, 148.68° | Storm tide gauge | November 1994 | - 2.81 |
| Mackay | -21.04°, 149.55° | Wave buoy | September 1975 | 34 |
| Mackay | -21.11°, 149.23° | Storm tide gauge | June 1975 | - 2.94 |
| Hay Point | -21.27°, 149.31° | Wave buoy | February 1993 | 10 |





**Table 2.** Maximum surge, storm tide and minimum atmospheric pressure recorded at gauges (north to south) during *Debbie*. Calculations of barometric surge and wind setup components of the maximum surge are given. The time of maximum storm tide coincided with the astronomical high tide (approx. 2 hrs before landfall), not the time of maximum surge (which occurred 1 – 2 hrs after landfall for the southern sites). The predicted astronomical tide height *at the time of maximum storm tide* is shown below.

| Site name | Max. surge (m) | Barometric surge (m) | Wind setup (m) | Predicted tide (m AHD) | Max. storm tide (m AHD) | Contribution of predicted tide (%) | Min. atmos. pressure (hPa) |
|---|---|---|---|---|---|---|---|
| Bowen | 0.52 | 0.44 | 0.08 | 1.46 | 1.80 | 81 % | 969 |
| Shute Harbour | 1.23 | 0.56 | 0.67 | 1.71 | 2.63 | 65 % | 957 |
| Laguna Quays | 2.66 | 0.39 | 2.27 | 2.45 | 4.40 | 56 % | 974 |
| Mackay | 1.12 | 0.21 | 0.91 | 2.99 | 3.70 | 81 % | 992 |





**Table 3.** Wave conditions observed (north to south) during *Debbie*. Both maximum wave heights, and wave conditions at the time of maximum storm tide are shown.

| Site name | Maximum during *Debbie* | | Time of maximum storm tide | | |
|---|---|---|---|---|---|
| | $H_s$ (m) | $H_{max}$ (m) | $H_s$ (m) | $T_p$ (s) | $MWD$ (°) |
| Abbot Point | 2.9 | 5.5 | 1.7 | 5.8 | 215 (SW) |
| Mackay[*] | 7.1 | 10.7 | 4.7 | 9.5 | 103 (ESE) |
| Hay Point | 3.6 | 6.6 | 3.5 | 8.1 | 64 (ENE) |

[*] Wave conditions at Mackay are extrapolated and not observed values.





**Table 4.** Maximum limits of inundation during *Debbie*, from field surveys (north to south) by RF/MQU and DSITI. Storm tide height is maximum recorded at Shute Harbour gauge for Hamilton Island, Mackay gauge for Seaforth and Laguna Quays gauge for all others. Shute Harbour is probably not a good representation of the storm tide level at Hamilton Island, but was the closest gauge during *Debbie*. Additional sites were surveyed by DSITI and are detailed in DSITI (2017).

| Site name | Area (Group) | Mean value (and range), m AHD | No. obs. points | Storm tide, m (AHD) | Waves effects (m)[*] |
|---|---|---|---|---|---|
| Hamilton Island | North-east facing beach (DSITI) | 5.73 (5.52 – 5.90) | 40 | 2.63 (46 %) | 3.13 (54 %) |
| Conway Beach | West end of beach (DSITI) | 5.15 (4.86 – 5.89) | 5 | 4.40 (85 %) | 0.75 (15 %) |
| | Central frontage (DSITI) | 5.07 (4.76 – 5.55) | 11 | 4.40 (87 %) | 0.67 (13 %) |
| Wilson Beach | Wilson Beach frontage (DSITI) | 5.15 | 1 | 4.40 (85 %) | 0.75 (15 %) |
| Midge Point | Midge Point beach (DSITI) | 5.23 (5.07 – 5.50) | 33 | 4.40 (84 %) | 0.83 (16 %) |
| | Midge Point beach (RF/MQU) | 5.21 (4.99 – 5.47) | 190 | 4.40 (85 %) | 0.81 (15 %) |
| Midgeton | North end of town (DSITI) | 4.30 (4.10 – 4.54) | 26 | 4.40 | |
| | Central frontage (DSITI) | 4.25 (4.13 – 4.37) | 15 | 4.40 | |
| Laguna Quays | Side of marina (DSITI) | 4.62 (4.40 – 4.77) | 27 | 4.40 (95 %) | 0.22 (5 %) |
| | Side of marina (RF/MQU) | 4.60 (4.19 – 5.01) | 165 | 4.40 (96 %) | 0.20 (4 %) |
| Seaforth | North end of beach (DSITI) | 3.76 (3.55-3.87) | 17 | 3.69 (98 %) | 0.07 (2 %) |
| | Central frontage (DSITI) | 5.30 (5.02 – 5.66) | 10 | 3.69 (70 %) | 1.61 (30 %) |
| | Central frontage (RF/MQU) | 5.20 (4.90 – 5.52) | 146 | 3.69 (71 %) | 1.51 (29 %) |
| | South/central frontage (RF/MQU) | 4.76 (4.10 – 5.06) | 172 | 3.69 (78 %) | 1.07 (22 %) |
| | South end of beach (DSITI) | 4.12 (3.89 – 4.25) | 17 | 3.69 (90 %) | 0.43 (10 %) |

5    [**] at each site, the residual between the mean of the maximum water level observations and the storm tide height from the nearest gauge was used to estimate the contribution of wave effects. At Midgeton, the storm tide is greater than the total water level estimate, for reasons described in the text.



**Table 5.** Beach erosion volumes for erosion zones referred to in Figure 7. Bracketed values show volume range within reported accuracy of LiDAR ($\pm$ 0.15 m to one RMSE, or 68% confidence).

| Erosion zone | Eroded beach volume (m$^3$) |
|---|---|
| CB 1 - defended beach | -2,512 (-1,597 to -3,484) |
| CB 2 - undefended beach | -960 (-378 to -1,579) |
| CB 3 - creek entrance | -3,677 (-2,726 to -4,723) |
| WB 1 - main beach | -4,457 (-3,075 to -5,939) |
| MB 1 - beach S of creek | -1,502 (-564 to -2,613) |
| MB 2 – S creek entrance | -3,681 (-2,213 to -5,363) |
| MB 3 - main beach | -12,457 (-9,610 to -15,812) |
| MB 4 - N creek entrance | -2,647 (-1,709 to -3,686) |



**Table 6.** Cross-shore beach profile changes above 2 m AHD. Bracketed values show percentage of eroded volume.

| Beach profile | Erosion ($m^3$ m) | Accretion ($m^3$ m) | Missing volume ($m^3$ m) |
|---|---|---|---|
| MP1 | -13.9 | +0.6 (4.3 %) | 13.3 (95.7 %) |
| MP2 | -19.3 | +3.6 (18.7 %) | 15.7 (81.3 %) |
| MP3 | -13.7 | +5.1 (37.2 %) | 8.6 (62.8 %) |
| CB1 | -8.8 | +1.6 (18.2 %) | 7.2 (81.8 %) |
| CB2[*] | -6.1 | +9.9 (162.3 %) | |
| CB3[*] | -7.1 | +9.5 (133.8 %) | |
| WB1 | -9.3 | +1.9 (20.4 %) | 7.4 (79.6 %) |
| WB2 | -8.6 | +2.5 (29.1 %) | 6.1 (70.9 %) |
| WB3 | -9.6 | +2.6 (27.1 %) | 7.0 (72.9 %) |

[*] All profiles saw a net loss of sand above 2 m AHD, except CB2 and CB3 which experienced a net gain (an additional 62 and 34 % of sand, respectively).