# Peer review of "Extreme water levels, waves and coastal impacts during a severe tropical cyclone in Northeast Australia: a case study for cross-sector data sharing"

_Natural Hazards and Earth System Sciences, 2018_

## Referee Comment (RC1) · Anonymous Referee #1 · 31 May 2018

General Comments: This manuscript brings together observations from diverse sources and analyzes these in the context of spatial variations in coastal impacts from tropical cyclone Debbie that made landfall on the eastern Australian coast in March 2017. Compiled data of waves, water levels, and beach erosion coupled with some relatively simple and straightforward analytical calculations of wave extrapolations, wave runup, and relative contributions of low pressures and wind-setup to storm surge, and observations are used to evaluate flooding, erosion, and infrastructure damage and makes causal connections of these patterns with duration of the storm, timing with as-

tronomic tides, and locations and geomorphological settings of the most and affected areas. The paper highlights the importance of data sharing across industry, government and academia to improve understanding and reduce coastal risk. The paper is very well written: it is clear, concise, and draws clear connections between the data, results, and conclusions, including pointing out the uncertainties. The figures and tables are clear and not superfluous nor lacking.

Specific Comments: The only comment I have is that it is stated throughout the paper that storm surge and tides are the largest contributor to the total water level; it appears that this is true but supporting evidence is not really presented. The Stockdon (and other) runup equation is very sensitive to foreshore slope. Whereas the foreshore slopes employed at each of the sites seem reasonable, tanB could have been quite different when TC Debbie made landfall, since the measurements were obtained 5 months prior. I recommend that a sensitivity test of the relative contribution of the wave runup, inclusive of a range of plausible foreshore slopes, to the total water level be included in the study.

---

## Author Comment (AC1) · 31 May 2018

The reviewer's comments relate to the contribution of wave effects to total water levels during TC Debbie, and the sensitivity of the parametric formulation of Stockdon et al. to beach foreshore slope in estimating wave runup contribution to coastal water levels.

Regarding the reviewer's first point: "it is stated throughout the paper that storm surge and tides are the largest contributor to the total water level; it appears that this is true but supporting evidence is not really presented."

[Figure]

The average astronomic tide (from observations of all the gauges in the area), at time of maximum water levels, was 2.15 m; the average surge residual, at the time of maximum water levels, was 0.98 m. In terms of waves, the contribution to maximum coastal water levels - in the absence of detailed modelling - can only be inferred through parametric estimates, or (even better) by subtracting the observed storm tide (tide + surge) from total coastal water levels inferred from field measurements. In this study, we did both, and used the latter as a check on the parametric estimates (see section 5.3).

We would argue that [total coastal water level observations – storm tide observations] is the best supporting evidence possible for the contribution of wave effects to total coastal water levels. Results from this approach suggested that, on average, wave effects contributed $\sim$ 16 % to water levels on the open coast. Conceptually, this fits with the wide, shallow shelf of this locality – which produces a large tidal range, is conductive to large surge events but dissipates wave energy.

Regarding the reviewer's second point: "Whereas the foreshore slopes employed at each of the sites seem reasonable, tanB could have been quite different when TC Debbie made landfall, since the measurements were obtained 5 months prior. I recommend that a sensitivity test of the relative contribution of the wave runup, inclusive of a range of plausible foreshore slopes, to the total water level be included in the study."

The Stockdon formula was chosen because of its development for cyclone-like conditions, and largely successful application in E Australia. The reviewer is quite correct, however, that it is simplistic and very sensitive to foreshore slope - and this is something that is discussed particularly in terms of the Wilson Beach site, where the upper and lower foreshore slopes are very different (see section 5.3). It is also true that the foreshore slope was obtained from LiDAR transects flown $\sim$ 5 months prior to Debbie and thus may have been different to when Debbie made landfall. This point is also highlighted in the paper (see sections 4.6).

However, our analysis of buoy data (section 5.2), indicates a very low wave energy

regime prevailed during the months preceding Debbie in this region, and likewise low wave energy, and very small beach profile change, was observed five months post Debbie (as discussed in section 5.6). Both indicate there is very little beach morphological change - particularly with regard to significant changes in beach foreshope slope - outside cyclonic events, at least on sub-annual timescales.

Therefore, we believe the reviewer's suggestion of using a range of plausible pre-Debbie foreshore slopes, while a useful exercise for wave-dominated coasts with a moderate modal wave energy climate, will not differ significantly from our pre-storm values at this location.

In writing the paper, we did consider using a range of upper and lower foreshore slopes, and the mean of the two, to derive runup estimates (as discussed briefly in section 5.3). This is because, as highlighted in Stockdon et al 2006 and 2007, the surf zone widens considerably during cyclone (or hurricane) events and thus the 'mean foreshore slope' as parameterized, is not necessarily equal to the upper high-tide beach slope.

However, we were unable to explore this because the lower foreshore slope was not captured entirely by the LiDAR. The LiDAR was flown for terrestrial purposes at mid-tide, thus not capturing the entire low-tide beach.

---

## Referee Comment (RC2) · Anonymous Referee #2 · 12 Jun 2018

This manuscript offers a picture of the impact of the tropical cyclone Debbie during its landfall in Northeast Australia, from the point of view of measurements. Waves and storm surge measurements come from buoys and tidal gauges along the shore. The contribution to the water levels of waves was estimated using the Stockdon empirical relation. The impact of the storm, in terms of erosion, was measured with lidar scans. Also the extent of the flooded area was checked, and found consistent with the estimated tide + surge + wave setup and runup.

The data presented here are scientifically relevant, as they cover this storm in quite a

comprehensive way, that is positive not only to improve local risk management. Modelling tropical cyclones and their impact is still a challenge also due to the lack of measurements, and detailed descriptions of single events can help improving our models.

Furthermore, this paper is well written. Therefore I recommend it for publication on NHESS.

I have only a few comments, listed here below

- Apart from what already done, it would be nice to have the ratio between the contribution of waves to water levels (setup + runup) and significant wave height. That's because in large scale studies, where an accurate measure of beach slope is not available, the wave contribution is sometimes taken as a fixed fraction of hs.

- Is there for Debbie any offshore measurement for waves and storm tide? Maybe from satellite altimeters? If the authors could provide some deep water measurement, along or close to the track, the set of data provided here would be really complete.

- Pag 5, line 20, how can the authors be sure that the tidal gauges don't measure some wave setup? This would be possible in such extreme conditions. It would lead to an overestimation of wind setup, and the application of Stockdon would lead to an overestimation of total water level.

- formula (2): I would suggest either to indicate with beta the slope, either to put tan beta in parenthesis.

- pag 6, line 33: is it possible that with such extreme Hs, at 35m depth you already significant wave breaking? In this case you would be underestimating Hs.

- formula 8: wouldn't it more proper to call this measure the "total energy released per unit coast length"

- pag 12, line 30: just as a matter of speculation, is it possible that the overestimation could be also due to a contribution of wave setup in the mesurements of some gauge,

then the authors would re-add this contribution estimating r2%?

- the figures are generally nice, but adding clear legends would make them more clear.

- figure 1: the meaning of the symbols and lines is clear only after reading the description. It would be nice to have a legend explaining what the dots and patterns are.

- figure 2: legend missing

- figure 3: the legend should explain also what the dashed lines represent

- figure 4: given the orientation of the coast, I believe it would be clearer to show A B and C in the right side, D in the left side of the figure (maybe write on the top East and West, rather than North and South)

- figures 5 and 6, I would write the name of the location in the figures after A B C and D

- figure 6, legend is missing

- figure 8: add legend, and write the name of the location close to the panel id

---

## Author Comment (AC2) · 22 Jun 2018

We thank the reviewer for their thoughtful review of our manuscript. We have addressed each of the reviewer's comments point-by-point below:

1. "...it would be nice to have the ratio between the contribution of waves to water levels (setup + runup) and significant wave height.".

This is a good point that we agree makes a useful addition to the paper. We have included this information as an extra column in Table 4, with a footnote for descrip-

tion. We have also included a short discussion on the findings, in the last paragraph in section 5.3, and the forth paragraph in the conclusions. An extra reference (Nott, 2003) has been added. We find wave runup to be, on average, 18 % of the offshore significant wave height which is consistent with previous studies. There was, however, large variation between sites based on wave exposure.

2. "Is there for Debbie any offshore measurement for waves and storm tide? Maybe from satellite altimeters?"

Satellite altimetry-derived wave height and sea level measurements are indeed available for the Australian region. However, we feel the addition of this information is outside the scope of the paper for two reasons. The first is that the focus of the paper is on hydrodynamic drivers and impacts in the coastal zone, and at this location, shallow to intermediate water depths extend across the wide GBR shelf to approximately 100 km offshore. Thus, the value added by including altimetric deep-water wave height data in terms of coastal impacts is questionable. The second is that a core aim of this paper is to demonstrate the power of data sharing in a post-disaster environment. All data analysed in this study has been collected and shared amongst the contributing partners and our preference is to limit the analysis to this. Even with this arbitrary limit, we believe it constitutes a substantial body of work.

3. "Pag 5, line 20, how can the authors be sure that the tidal gauges don't measure some wave setup?"

This is a pertinent point, and something that was mentioned during internal review but appears to have not been discussed in the text. The four tide gauges used in this study (Mackay, Laguna Quays, Shute Harbour and Bowen) are installed by Queensland Government sufficiently far outside the wave breaking zone to not include wave setup, under normal circumstances. They are also all installed on pier or wharf locations which are typically sheltered from wave breaking by design. However, it is true that under extreme conditions there may be a small component of wave setup that is captured

in a time-averaged sense at the gauges. Our knowledge of these locations makes us believe the contribution would be minimal. We have now updated the text to include consideration of this point, as an additional paragraph in section 3.1.

4. "formula (2): I would suggest either to indicate with beta the slope, either to put tan beta in parenthesis."

Equation 2 has now been updated accordingly.

5. "pag 6, line 33: is it possible that with such extreme Hs, at 35m depth you already significant wave breaking?"

We believe not. Wave breaking usually occurs at a height-to-depth ratio of approximately 0.7, meaning that even the highest Hmax value inferred in this study (approx. 10 m) would begin to break in around 14 m of water depth. This is safely shoreward of the Mackay buoy in 35 m water depth. For wave breaking to occur at the shallower buoy locations (e.g. Hay Point at 10 m water depth) waves would need to be 7 m or greater. However, only wave heights < 4 m were recorded at this location – suggesting all buoy data represents a shoaled but unbroken wave climate.

Words to this effect have been added in section 3.3 and as an additional paragraph in section 5.2 to clarify this for the reader.

6. "formula 8: wouldn't it more proper to call this measure the "total energy released per unit coast length"

This definition has been added to the text below Eq. 8.

7. "pag 12, line 30: just as a matter of speculation, is it possible that the overestimation could be also due to a contribution of wave setup in the measurements of some gauges"

The consideration of wave setup inclusion in the storm tide gauge data has been addressed in response to reviewer's comment #3. The potential effect of this with regard

to calculation of R2% we believe to be negligible because of all the gauges analysed, the most sheltered of these (Laguna Quays) was used as a surrogate storm tide measurement for the three beach study sites. Inaccuracies in the parameterization of the beach slope would have a far larger effect on R2%, as we have discussed in the paper.

8. "figure 1: . . . It would be nice to have a legend explaining what the dots and patterns are".

A legend has now been added to figure 1.

9. "figure 2: legend missing"

Legends have now been added to figure 2.

10. "figure 3: the legend should explain also what the dashed lines represent"

Legend has been extended to include dashed lines.

11. "figure 4: given the orientation of the coast, I believe it would be clearer to show A B and C in the right side, D in the left side of the figure (maybe write on the top East and West, rather than North and South)"

The orientation of the coast is North-South, not East-West. Gauges 'A' to 'C' are indeed south of the cyclone eye and 'D' is north of the eye. Thus, the labelling and sequence is correct here.

12. "figures 5 and 6, I would write the name of the location in the figures after A B C and D"

Names have been added to A B C D plate titles in Figure 5, and A B C plates in Figure 6.

13. "figure 6, legend is missing"

No space for legend in Figure 6, so each item has been annotated on the plot to fully describe meaning.

14. "figure 8: add legend, and write the name of the location close to the panel id"

Legend and location names have been added to Figure 8.

All these changes have been actioned and will appear in the final version of the manuscript.

---

## Author Response (AR2)

**Author responses to Reviewer #1 and #2 comments (second round)**

Thank you for reviewing the revised version of our manuscript. The legend on Figure 2 (panels A and D) have now been added.

---

## Author Response (AR3)

**Author responses to Reviewer #1 comments**

We thank the reviewer for their thoughtful review of our manuscript. The reviewer's comments relate to the contribution of wave effects to total water levels during TC *Debbie*, and the sensitivity of the parametric formulation of Stockdon et al. to beach foreshore slope in estimating wave runup contribution to coastal water levels.
* * *
Regarding the reviewer's first point: *"it is stated throughout the paper that storm surge and tides are the largest contributor to the total water level; it appears that this is true but supporting evidence is not really presented."*

The average astronomic tide (from observations of all the gauges in the area), at time of maximum water levels, was 2.15 m; the average surge residual, at the time of maximum water levels, was 0.98 m. In terms of waves, the contribution to maximum coastal water levels - in the absence of detailed modelling - can only be inferred through parametric estimates, or (even better) by subtracting the observed storm tide (tide + surge) from total coastal water levels inferred from field measurements. In this study, we did both, and used the latter as a check on the parametric estimates (see section 5.3).

We would argue that [total coastal water level observations – storm tide observations] is the best supporting evidence possible for the contribution of wave effects to total coastal water levels. Results from this approach suggested that, on average, wave effects contributed ~ 16 % to water levels on the open coast. Conceptually, this fits with the wide, shallow shelf of this locality – which produces a large tidal range, is conductive to large surge events but dissipates wave energy.
* * *
Regarding the reviewer's second point: "*Whereas the foreshore slopes employed at each of the sites seem reasonable, tanB could have been quite different when TC Debbie made landfall, since the measurements were obtained 5 months prior. I recommend that a sensitivity test of the relative contribution of the wave runup, inclusive of a range of plausible foreshore slopes, to the total water level be included in the study.*"

The Stockdon formula was chosen because of its development for cyclone-like conditions, and largely successful application in E Australia. The reviewer is quite correct, however, that it is simplistic and very sensitive to foreshore slope - and this is something that is discussed particularly in terms of the Wilson Beach site, where the upper and lower foreshore slopes are very different (see section 5.3).

It is also true that the foreshore slope was obtained from LiDAR transects flown ~ 5 months prior to *Debbie* and thus may have been different to when *Debbie* made landfall. This point is also highlighted in the paper (see sections 4.6).

However, our analysis of buoy data (section 5.2), indicates a very low wave energy regime prevailed during the months preceding *Debbie* in this region, and likewise low wave energy, and

very small beach profile change, was observed five months post *Debbie* (as discussed in section 5.6). Both indicate there is very little beach morphologic change - particularly with regard to significant changes in beach foreshope slope - outside cyclonic events, at least on sub-annual timescales.

Therefore, we believe the reviewer's suggestion of using a range of plausible pre-*Debbie* foreshore slopes, while a useful exercise for wave-dominated coasts with a moderate modal wave energy climate, is unlikely to differ significant from our pre-storm values at this location.

In writing the paper, we did consider using a range of upper and lower foreshore slopes, and the mean of the two, to derive runup estimates (as discussed briefly in section 5.3). This is because, as highlighted in Stockdon et al 2006 and 2007, the surf zone widens considerably during cyclone (or hurricane) events and thus the 'mean foreshore slope' as parameterized, is not necessarily equal to the upper high-tide beach slope.

However, we were unable to explore this because the lower foreshore slope was not captured entirely by the LiDAR. The LiDAR was flown for terrestrial purposes at mid-tide, thus not capturing the entire low-tide beach.

**Author responses to Reviewer #2 comments**

We thank the reviewer for their thoughtful review of our manuscript. We have addressed each of the reviewer's comments point-by-point below:

1. "*…it would be nice to have the ratio between the contribution of waves to water levels (setup + runup) and significant wave height*.".

This is a good point that we agree makes a useful addition to the paper. We have included this information as an extra column in Table 4, with a footnote for description. We have also included a short discussion on the findings, in the last paragraph in section 5.3, and the fourth paragraph in the conclusions. An extra reference (Nott, 2003) has been added. We find wave runup to be, on average, 18 % of the offshore significant wave height which is consistent with previous studies. There was, however, large variation between sites based on wave exposure.

2. "*Is there for Debbie any offshore measurement for waves and storm tide? Maybe from satellite altimeters?*"

Satellite altimetry-derived wave height and sea level measurements are indeed available for the Australian region. However, we feel the addition of this information is outside the scope of the paper for two reasons. The first is that the focus of the paper is on hydrodynamic drivers and impacts in the coastal zone, and at this location, shallow to intermediate water depths extend across the wide GBR shelf to approximately 100 km offshore. Thus, the value added by including altimetric deep-water wave height data in terms of coastal impacts is questionable. The second is that a core aim of this paper is to demonstrate the power of data sharing in a post-disaster environment. All data analysed in this study has been collected and shared amongst the

contributing partners and our preference is to limit the analysis to this. Even with this arbitrary limit, we believe it constitutes a substantial body of work.

3. "*Pag 5, line 20, how can the authors be sure that the tidal gauges don't measure some wave setup?*"

This is a pertinent point, and something that was mentioned during internal review but appears to have not been discussed in the text. The four tide gauges used in this study (Mackay, Laguna Quays, Shute Harbour and Bowen) are installed by Queensland Government sufficiently far outside the wave breaking zone to not include wave setup, under normal circumstances. They are also all installed on pier or wharf locations which are typically sheltered from wave breaking by design. However, it is true that under extreme conditions there may be a small component of wave setup that is captured in a time-averaged sense at the gauges. Our knowledge of these locations makes us believe the contribution would be minimal. We have now updated the text to include consideration of this point, as an additional paragraph in section 3.1.

4. "*formula (2): I would suggest either to indicate with beta the slope, either to put tan beta in parenthesis.*"

Equation 2 has now been updated accordingly.

5. "*pag 6, line 33: is it possible that with such extreme Hs, at 35m depth you already significant wave breaking?*"

We believe not. Wave breaking usually occurs at a height-to-depth ratio of approximately 0.7, meaning that even the highest $H_{max}$ value inferred in this study (approx. 10 m) would begin to break in around 14 m of water depth. This is safely shoreward of the Mackay buoy in 35 m water depth. For wave breaking to occur at the shallower buoy locations (e.g. Hay Point at 10 m water depth) waves would need to be 7 m or greater. However, only wave heights < 4 m were recorded at this location – suggesting all buoy data represents a shoaled but unbroken wave climate.

Words to this effect have been added in section 3.3 and as an additional paragraph in section 5.2 to clarify this for the reader.

6. "*formula 8: wouldn't it more proper to call this measure the "total energy released per unit coast length*"

This definition has been added to the text below Eq. 8.

7. "*pag 12, line 30: just as a matter of speculation, is it possible that the overestimation could be also due to a contribution of wave setup in the measurements of some gauges*"

The consideration of wave setup inclusion in the storm tide gauge data has been addressed in response to reviewer's comment #3. The potential effect of this with regard to calculation of R2% we believe to be negligible because of all the gauges analysed, the most sheltered of these (Laguna Quays) was used as a surrogate storm tide measurement for the three beach study sites. Inaccuracies in the parameterization of the beach slope would have a far larger effect on R2%, as we have discussed in the paper.

8. "*figure 1: … It would be nice to have a legend explaining what the dots and patterns are*".

A legend has now been added to figure 1.

9. "*figure 2: legend missing*"

Legends have now been added to figure 2.

10. "*figure 3: the legend should explain also what the dashed lines represent*"

Legend has been extended to include dashed lines.

11. "*figure 4: given the orientation of the coast, I believe it would be clearer to show A B and C in the right side, D in the left side of the figure (maybe write on the top East and West, rather than North and South)*"

The orientation of the coast is North-South, not East-West. Gauges 'A' to 'C' are indeed south of the cyclone eye and 'D' is north of the eye. Thus, the labelling and sequence is correct here.

12. "*figures 5 and 6, I would write the name of the location in the figures after A B C and D*"

Names have been added to A B C D plate titles in Figure 5, and A B C plates in Figure 6.

13. "*figure 6, legend is missing*"

No space for legend in Figure 6, so each item has been annotated on the plot to fully describe meaning.

14. "*figure 8: add legend, and write the name of the location close to the panel id*"

Legend and location names have been added to Figure 8.

**Author responses to Reviewer #1 and #2 comments (second round)**

Thank you for reviewing the revised version of our manuscript. The legend on Figure 2 (panels A and D) have now been added.

[revised manuscript text omitted]

**Commented [TM10]:** REVIEWER #2 COMMENT 9
Legend added to figure 2

REVIEWER #2 COMMENT (round two)
Legend added to figure 2 panels A and D also

[Figure]

**Figure 3:** Water levels (A) and storm surge residuals (B) observed at (north to south) Bowen, Shute Harbour, Laguna Quays and Mackay storm tide gauges during TC *Debbie*. Highest Astronomical Tide (HAT, dotted horizontal lines), times of maximum water level (circles in A) and times of maximum surge residuals (circles in B) are also shown. Red vertical line indicates approximate time of landfall at Airlie Beach. Time is in Australian Eastern Standard Time (AEST, UTC + 10 hrs). Track of *Debbie* during this period (up to landfall) can be seen in Figure 1.

**Commented [TM11]:** REVIEWER #2 COMMENT 10
Legend has been extended to include dashed lines

[Figure]

**Figure 4:** Maximum storm surge and components (barometric surge, blue, and wind setup, orange) plotted as a function of distance from *Debbie* landfall at 12:00 on 28 March 2017 (solid red line), where A = Mackay, B = Laguna Quays, C = Shute Harbour and D = Bowen storm tide gauge stations (black lines show centroids of each). Radius of maximum winds (RMW, red dotted lines), radius of hurricane-force winds (RHFW, black dotted lines) are also shown. The maximum surge north and south of *Debbie's* eye (black curve and circles) demonstrates the asymmetry in the surge, skewed to the south of the cyclone centre.

[Figure]

**Figure 5:** Wave observations during *Debbie* at (north to south) Abbot Point, Mackay and Hay Point wave buoys. Parameters shown are (A) significant wave height, $H_s$, (B) maximum wave height, $H_{max}$, (C) peak wave period, $T_p$ and (D) mean wave direction, *MWD*. Red vertical line indicates approximate time of landfall at Airlie Beach. Dotted blue line indicates the extrapolated estimate of wave conditions at Mackay, after 5 the failure of the wave buoy at 05:30 on 28 March. Maximum $H_s$ and $H_{max}$ at each buoy are also shown in (A) and (B), respectively (circles).

**Commented [TM12]:** REVIEWER #2 COMMENT 12
Names have been added to A B C D plate titles in Figure 5

[Figure]

**Figure 6:** Pre-*Debbie* beach profiles (July-August 2016) at (A) Midgeton Beach, (B) Conway Beach and (C) Wilson Beach (profile locations in Figure 7). Maximum storm tide elevation was taken from the nearest gauge (Laguna Quays) (grey solid line), and maximum water elevation (grey dashed line) was defined as $R_{2\%}$ + storm tide elevation. $R_{2\%}$ corresponds to the expected level of exceedance of 2% of the wave runups during wave conditions at the time of maximum storm tide elevation ($H_{s0}$ 5 m $L_0$ 131 m, at 12:00 28 Mar). $R_{2\%}$ was calculated for each location based on the mean slope ($\tan\beta$) of the profiles. The locations at which $\tan\beta$ was calculated for each profile are shown (base of dune to toe of upper beach, black circles).

**Commented [TM13]:** REVIEWER #2 COMMENT 12 and 13
Names have been added to A B C plate titles in Figure 6

Each item now annotated in the plots to fully describe meaning

[revised manuscript text omitted]